# Linearly Controlled Language Generation with Performative Guarantees

## Abstract

The increasing prevalence of Large Language Models (LMs) in critical applications highlights the need for controlled language generation strategies that are not only computationally efficient but that also enjoy performance guarantees. To achieve this, we use a common model of concept semantics as linearly represented in an LM's latent space. In particular, we take the view that natural language generation traces a trajectory in this continuous semantic space, realized by the language model's hidden activations. This view permits a control-theoretic treatment of text generation in latent space, in which we propose a lightweight, gradient-free intervention that dynamically steers trajectories away from regions corresponding to undesired meanings. Crucially, we show that this intervention, which we compute in closed form, is guaranteed (in probability) to steer the output into the allowed region. Finally, we demonstrate on a toxicity avoidance objective that the intervention steers language away from undesired content while maintaining text quality.

WARNING: This paper contains model outputs which are offensive in nature.

## 1 Introduction

Language Models (LMs) have become widespread in critical applications such as content moderation and real-time information dissemination (Zeng et al., 2024). Despite their transformative impact, these models require updates to remain accurate post-deployment. Moreover, as demand for more nuanced text generation rises, strategies that enforce constraints during text generation are increasingly needed. To address these challenges, controllable text generation has emerged as a pivotal research area.

Several approaches have been proposed towards controllable text generation (Kumar et al., 2021; Lu et al., 2021; Li et al., 2022; Qin et al., 2022). Of them, a popular approach is prompt engineering (Luo et al., 2023; Bhargava et al., 2023; Cai et al., 2023), where natural language prompts are carefully chosen at input-time to steer generation. Other approaches modify LM parameters to achieve the desired outputs (Yao et al., 2023; Li et al., 2023b). Lastly, some approaches engineer LM *activations*, or input representations, to steer them into the representations of desired outputs (Dathathri et al., 2019; Hernandez et al., 2023; Konen et al., 2024; Li et al., 2024a).

Despite current efforts, ensuring the controllability of these models remains a challenge due to their limited interpretability. For instance, while knowledge editing provides an efficient alternative to exhaustive retraining, it poses risks akin to the butterfly effect: minor adjustments could lead to unintended consequences. Moreover, it is paramount for these approaches to be robust and ensure controllability guarantees to mitigate risks and harness their full potential safely.

To address this gap, we propose to use control theory to tackle controlled language generation. Specifically, optimal control theory (Kirk, 2004) offers principled methods to steer trajectories in latent space that enjoy theoretical guarantees on the performance of the intervention. In the framework of optimal control theory, our intervention method, which we call Linear Semantic Control (LiSeCo), derives from a theoretical formulation of controlled text generation. Our contributions are both theoretical and empirical: (1) we formally pose LM control as a constrained optimization problem and provide its closed-form solution with guarantees; (2) we empirically demonstrate our method on the use cases of toxicity and negativity avoidance. We confirm, with experiment corroborating theory, that LiSeCo indeed steers LM generation from disallowed concepts while maintaining text quality.

## 2 RELATED WORK

Contemporary language models are deep neural networks pre-trained on trillions of tokens of Internet-scale text. In part due to their vast scale and lack of interpretability, methods to control them in a fine-grained way remain elusive. A number of approaches have already been proposed towards this end, spanning the whole spectrum of permanent (Meng et al., 2022b; Belrose et al., 2023) to online control strategies (Liu et al., 2021; Dathathri et al., 2019). Here, we review post-hoc intervention methods and situate LiSeCo with respect to the current landscape.

Post-hoc intervention methods can intervene on various components of the LM: for instance, decoding, like FUDGE and GeDI (Yang & Klein, 2021; Krause et al., 2021), activations, like LiSeCo, or weights via finetuning. All such methods aim to modify some attribute, such as toxicity, while maintaining text fluency. Ultimately, all methods work towards this goal by modifying the LM's final probability distribution, either directly or indirectly. We can situate where different method classes intervene, viewing an LM as a series of $T$ function compositions corresponding to the $T$ layers, where $s$ is a sequence of tokens:

$$\mathbb{P}_{LM}(s_i|s_{<i}) = f_T \circ f_{T-1} \cdots \circ f_1(s_{<i}) := LM(s_{<i}).$$

**Decoding-based methods** fix the function $LM := f_T \circ f_{T-1} \circ \cdots f_1$ and directly edit its output probability distribution $\mathbb{P}_{LM}(s_i|s_{<i})$ (Yang & Klein, 2021; Liu et al., 2021; Krause et al., 2021). These methods require access to an external evaluator whose feedback is used to calibrate token probabilities, which can result in high inference latency.

**Prompt engineering** is a technique that controls the LM's output by varying the input context $s_{<i}$, keeping the function $LM := f_T \circ f_{T-1} \circ \cdots f_1$ fixed (Luo et al., 2023; Bhargava et al., 2023; Cai et al., 2023; Wei et al., 2022; Li & Liang, 2021). Prompts are often highly task-specific, requiring either manually crafting or ad-hoc computationally-taxing techniques, and success can be brittle to prompt choice (Weber et al., 2023). While the space of natural language prompts is discrete, LM weights and activations live in continuous high-dimensional space, which is more expressive; then, rather than search over discrete prompts, other approaches that exploit this expressivity directly intervene in the internals of the model.

Of them, **weight-based methods** modify the functions $f_i$ themselves, which permanently constrains the space of final probability distributions $\mathbb{P}_{LM}$. These methods comprise, e.g., reinforcement learning from human feedback (Ouyang et al., 2022), instruction-tuning, parameter-efficient adaptation (Hu et al., 2022), or targeted weight-editing (Meng et al., 2022b; Belrose et al., 2023). In such approaches, weights are modified according to the goal of the controlled generation by, for instance, learning the necessary update (De Cao et al., 2021; Mitchell et al., 2021), or localizing and editing target parameters encoding specific knowledge (Dai et al., 2022; Meng et al., 2022a;c; Li et al., 2024b). Pitfalls range from potential inconsistencies and distortions, to the fact that weight-based methods can only correct errors in the LM's parametric knowledge, but not in-context (Li et al., 2023b).

**Activation-based methods**, such as LiSeCo, fix $LM := f_T \circ f_{T-1} \circ \cdots f_1$, but intervene at the domain of each $f_i$, where introducing a steering vector transforms the input to $f_i$ (Li et al., 2023a; Turner et al., 2023). These interventions can be seen as restricting the domain of each $f_i$, eventually constraining the space of probability distributions $\mathbb{P}_{LM}$ when composed up through the layers. A key advantage of activation steering is rapid adaptation that is *context-dependent*. An initial work in this domain was Plug and Play (Dathathri et al., 2019), where a linear intervention is computed at every layer. The control goal is encoded as the objective function in an optimization that is then solved via back-propagation, adding significant computational overhead at inference time. Subsequent approaches also compute linear modifications to the latent state, but reduce computational overhead, act on only a few layers (Subramani et al., 2022; Konen et al., 2024), or pre-compute steering vectors to avoid back-propagation (Turner et al., 2023). The recent approach REMEDI (Hernandez et al., 2023) also finds an optimal intervention to achieve different target outputs, but, requires specific a-priori training data for the representations, and offers no guarantees that the intervention will attain the target output. The approach presented in ITI (Li et al., 2024a) addresses the issue of computational efficiency at the expense of optimality (the intervention is not formulated as an optimizer), and lacks guarantees on the intervention's performance.

The use of steering vectors for text generation proposed in the literature provides empirical grounding for the promise of this approach. However, none of the approaches found in the literature provide

guarantees on the controllability of their method. Here, we provide an intervention that is *theoretically guaranteed* to steer the input into the allowed region, and introduces minimal computational overhead. Our work differs from existing literature by the following novel contributions:

1. We formally derive the optimal steering vector and offer theoretical performance guarantees. This is enabled by the use of an optimal control framework to cast the problem for the first time in a domain that has been overwhelmingly empirical.

2. Deriving the closed-form solution for the intervention minimizes increased latency at inference time. We provide empirical comparisons, where we show that other popular methods, such as FUDGE, require much higher latency.

Though Soatto et al. (2023) apply theoretical tools from control to LM text generation, to the best of our knowledge, our method is the first to propose a control-theoretic intervention whose theoretical guarantees are validated in practice.

## 3 PROBLEM STATEMENT

In this section we present the problem studied in this paper, as well as the assumptions and approach. In particular, we approach the problem of controlled language generation as a standard optimal control problem in the field of control theory (Kirk, 2004).

### 3.1 PROBLEM FORMULATION

Given a language model $\ell : \Sigma^* \to \Sigma^*$, controlled language generation aims to steer the model's output into a desired one. In this work, we study how to steer the output of an *already trained model* away from a disallowed region, or so that the set of possible generated sequences is constrained to an allowed subset $\mathcal{S} \subset \Sigma^*$. The requirements for the generated output sequence are two fold: its latent trajectory (1) is *guaranteed* to never lie in the disallowed region, and (2) stays as close as possible to that of the original output sequence, so that text quality is not compromised. In doing this, two questions need to be answered:

1. How can the disallowed region be defined for a given language model?
2. How can an intervention be designed to be guaranteed (in probability) to stay within the allowed region while retaining maximal similarity with the original model?

In what follows, we provide an answer to these questions, and show that the proposed approach adds minimal computational overhead to the language generation process.

### 3.2 APPROACH

We design an online method that guarantees, at each token generation, that the sequence remains safe. Given the sequential, feedforward nature of LM layers, we consider each new token generation to realize a trajectory through the layers' activation spaces. Similar to Park et al. (2023), we take the view that disallowed language occupies a region of each layer's activation space $\mathbb{R}^d$. Formally, let $\mathcal{R}_t \subset \mathbb{R}^d$ be the forbidden, or unsafe, region for layer $t$ of the LM. Our goal, then, is to provide a control mechanism by altering the output vector embedding at every layer that guarantees (in probability) that latent trajectories remain out of $\mathcal{R}_t$ for all $t$, see Figure 1. For each token generation pass, this control mechanism is to be applied at each layer. This control intervention is designed online, and it depends on the prompt sequence.

**Semantic Probe** We first identify the disallowed region for the generated token, given context. To do so, we feed a set of sequences to the model, and use a lightweight probe to map each latent state $x_t \in \mathbb{R}^d$ to a probability that the sequence is toxic. Specifically, we rely on a *probing classifier function* $f_t$ that maps the latent space $\mathbb{R}^d$ to the decision space $[0, 1]$. For simplicity, we take $f_t$ to be a logistic regression classifier realized by a linear probe (Hewitt & Manning, 2019). Formally, $f_t : \mathbb{R}^d \to [0, 1]; x_t \mapsto \sigma(W_t^\top x_t)$, where $W_t \in \mathbb{R}^{d \times 2}$ and $\sigma : \mathbb{R}^2 \to \mathbb{R}$ is the sigmoid. For each layer, we define the disallowed region $\mathcal{R}_t$ to be the pre-image of an *unsafe* classification under $f_t$, using a predefined probability threshold $p$. That is, $\mathcal{R}_t := \{x |\ \sigma(W_t^\top x) \geq p\}$, where $p \in [0, 1]$.

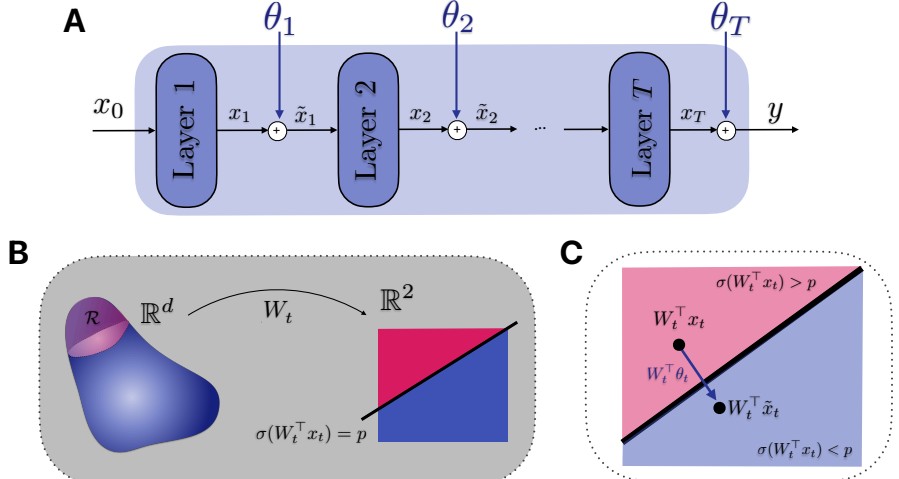

Figure 1: **A.** LiSeCo is based on linearly adding vector $\theta_t$ to the output of each layer ($t = 1, \ldots, T$). Each vector $\theta_t$ is the solution of a constrained optimization problem. **B.** A probing classifier $f$ mapping the latent space $\mathbb{R}^d$ to the decision space $[0, 1]$ is trained to characterize the allowable region (in blue) to which each latent state $\tilde{x}_t$ is constrained. Keeping trajectories out of the toxic region in latent space is equivalent to keeping their image out of the toxic region in decision space. **C.** At inference time, the state in latent space ($x_t \in \mathbb{R}^d$) is mapped via learned weights $W_t$ into $W_t^\top x_t \in \mathbb{R}^2$. If it falls within the forbidden region (pink), an intervention ($\theta_t \in \mathbb{R}^d$) is computed so that the updated state $\tilde{x}_t = x_t + \theta_t \in \mathbb{R}^d$ lies in the safe region (blue).

**Optimal Control** Once the forbidden region $\mathcal{R}_t$ is identified, we design a control strategy that, for all layers $t$, guarantees the latent state $x_t$ remains in the allowed region and retains maximal similarity with the original model. To do this, we design an optimal controller that generates an input $\theta_t \in \mathbb{R}^d$ at every layer $t$. Mathematically, we solve an optimization problem over $\theta_t$ where the pre-computed classifier enters as a hard constraint in the formulation, i.e., $\sigma(W_t^\top (\theta_t + x_t)) \le p$. This ensures the controlled latent trajectory $\tilde{x}_t = x_t + \theta_t \in \mathbb{R}^d$ lies in the unsafe region with probability less than $p$.

## 4 OPTIMAL LANGUAGE GENERATION CONTROLLER IN LATENT SPACE

In this section, we describe the theoretical contribution of this work. Using the probing classifier, we design a controller to restrict text generation to the safe region. The optimal intervention is derived in closed form, thus computationally efficient at inference-time.

### 4.1 OPTIMAL CONTROLLER SETUP

The optimal controller aims to keep latent trajectories out of the unsafe region without compromising text quality. That is, we perform constrained optimization where latent trajectories maximally approximate the original ones (proxying text quality) while avoiding the unsafe region as defined by the probe. This gives rise to the following optimization problem:

$$\min_{\theta_1, \ldots, \theta_T} \quad \sum_{t=1}^{T} \|\theta_t\|_2^2 \tag{1a}$$

$$s.t. \quad \sigma(W_t^\top (x_t + \theta_t)) - p \le 0, \quad \forall t = 1, \ldots, T \tag{1b}$$

$$x_{t+1} = \texttt{layer}_t(x_t + \theta_t), \tag{1c}$$

$$x_0 = E(\texttt{prompt sequence}), \tag{1d}$$

where $E$ is the embedding map. Optimization problem 1 aims to find the minimum $l_2$-norm intervention[1] $\theta_1, \ldots, \theta_T$ (Eq. 1a) that satisfies the following constraints: Eq. 1b requires the modified

---

[1]The choice of $L_2$ norm is standard in classical optimal control problems. The $L_2$ norm is usually interpreted as energy, or effort, of the control input to steer the system. In this context, it can be seeing as trying to minimize

latent state $x_t + \theta_t$ be classified unsafe by the probe $f_t$; Eq. 1c captures LM dynamics, i.e., layer $t$ maps the modified latent state $x_t + \theta_t$ to the next latent state $x_{t+1}$; Eq. 1d states that the LM's input embeds the input context, so that interventions are *context-dependent*. The intervention that solves optimization problem 1 is *guaranteed by construction* to keep the latent trajectory $\tilde{x}_1, \ldots, \tilde{x}_T$ and output $y$ below the probability threshold from the classifier.

Whether concept avoidance, e.g., detoxification, is expressed as a cost or a constraint depends on the use-case. Other approaches, in contrast to ours, encode concept avoidance in the optimization objective, but not via hard constraints (Dathathri et al., 2019; Hernandez et al., 2023). Fortunately, the constrained optimization framework of LiSeCo also permits this interpretation; though we leave its testing to future work, we state its equivalent problem and prove its *closed-form optimal solution*, which has only been empirically approximated by hyperparameter search in the literature (Li et al., 2023a), in Appendix D.

## 4.2 OPTIMAL CONTROLLER DESIGN

Optimization problem 1 is a standard problem in the optimal control literature (Kirk, 2004). By Bellman's Optimality Principle, the standard approach to solving problem 1 is dynamic programming (DP) (Kirk, 2004): the optimal solution is computed for the last layer $T$, then via backward induction for $T-1, \ldots, 1$. But, layer dynamics 1c are highly non-convex, and solutions incomputable in closed form, hence their optimality is not guaranteed. Further, DP requires backpropagating gradients at each text generation's forward pass, adding significant inference latency.

To overcome these limitations, we relax problem 1. No longer searching for a globally optimal solution across layers, we now search for locally optimal solutions at each layer. Now, Eqs. 1c and 1d cease to play a role, as each layer is optimized for separately. Then, problem 1 is relaxed into

$$\min_{\theta_t} \quad \|\theta_t\|_2^2 \tag{2a}$$

$$s.t. \quad \sigma(W_t^\top (x_t + \theta_t)) - p \leq 0, \tag{2b}$$

for each layer $t = 1 \cdots T$. The sequence of $\theta_t$ that solve problem 2 may not optimize the original formulation 1. However, one is not anyway guaranteed to find global optima anyway due to the high nonconvexity of layer computations. Furthermore, optimality is not essential as the cost aims only to preserve similarity with the original model. Meanwhile, the guarantee to avoid unsafe region $\mathcal{R}_t$ is still enforced via Eq. 2b.

A key advantage of relaxed formulation 2 is that it is solvable in closed-form, per-layer, with minimal computational overhead. The following theorem states the analytical solution for optimal $\theta_t$.

**Theorem 1** (Optimal $\theta$). *The optimal solution $\theta_t^* \in \mathbb{R}^d$ to the optimization problem 2 is given by*

$$\theta_t^* = \begin{cases} \dfrac{\log\left(\frac{1}{p} - 1\right) - w_t^\top x_t}{\|w_t\|_2^2} w_t & \text{if } \sigma(W_t^\top x_t) > p \tag{3a} \\ \\ 0 & \text{otherwise,} \tag{3b} \end{cases}$$

*where $w_t := W_t^1 - W_t^2$, the difference of the columns of $W_t =: \begin{bmatrix} W_t^1 & W_t^2 \end{bmatrix}$.*

*Proof.* Proof relies on leveraging the KKT conditions. See Appendix C for details. $\qquad\square$

Geometrically, the optimal solution is the vector from $x_t$ to the closest point in $\mathcal{R}_t^C$, which is the set-complement of $\mathcal{R}_t$. When $x_t \notin \mathcal{R}_t$, i.e., when $\mathbb{P}(\text{unsafe}) < p$,[2] no update is needed; hence $\theta_t^* = 0$. Otherwise, the update is a factor of $w_t$. Since $\theta_t^*$ exists in closed-form, computing an intervention incurs negligible computational overhead. Crucially, it is guaranteed with probability $p$ to keep the latent state outside the disallowed region.

---

the "effort" of the intervention. Moreover, the fact that $L_2$ also used to measure distances in an Euclidean space, like the embedding space that we consider in this work, makes it an appropriate choice to measure the "similarity" between the intervened representation and the original one.

[2] $\mathbb{P}$ denotes probability.

Although control occurs *locally* at each layer, the local control steps result in a globally safe distribution over the next token. To see this, consider a single token generation. Each sequential control action at layer $t$ guarantees that latent state $\tilde{x}_t \in \mathcal{R}_t^C$ is classified safe, or equivalently, eliminates the set of unsafe trajectories falling in $\mathcal{R}_t$. By the time we reach the last layer $T$, the latent trajectory is guaranteed (in-probability) to have been rated safe at every preceding layer. Then, the last layer $T$'s activation is transformed via the unembedding matrix (linear map) and softmax (monotonic map) to the distribution over the vocabulary. Linearity and monotonicity ensure that the set of safe last-layer activations maps onto the set of safe vocabulary distributions.

## 5 METHODS

The LiSeCo pipeline is as follows. First, to find the unsafe regions and probes per layer, there is an initial probe training phase. Then, probes are integrated into the model at inference-time, and the optimal intervention dynamically applied. We tested LiSeCo on two separate tasks: toxicity avoidance and negativity avoidance. For brevity, when appropriate, we will use *toxicity* to stand in for both toxicity and negativity in this section.

**Models** We test on three state-of-the-art causal language models: Llama-3-8B (Meta, 2024), Pythia-6.9B (Biderman et al., 2023), and Mistral-7B (Jiang et al., 2023). While the architectural details of a layer (attention + MLP) differ slightly between models, our intervention treats layers as black boxes and operates at the level of the *residual stream* (Elhage et al., 2021). This permits our intervention to be applied as a lightweight layer wrapper and in an architecture-agnostic way.

**Datasets** We test our method on the toxicity and negativity use cases. Borrowing terminology from Ashok & Poczos (2024), we first learn probing classifiers $f$ using a labelled *constraint dataset*, then, we evaluate text generation on a *task dataset*.

For **toxicity**, we use Kaggle's Jigsaw dataset (Adams et al., 2017) as the constraint dataset. The dataset contains 30k label-balanced natural language comments and their human-annotated binary toxicity labels in $\{\text{toxic}, \text{nontoxic}\}$. For the task dataset, we use RealToxicityPrompts, a dataset derived from OpenWebText, a large-scale corpus of the web (Gehman et al., 2020). RealToxicityPrompts is a collection of prompts, their continuations, and toxicity scores in $[0, 1]$ (Gokaslan & Cohen, 2019). To form the task dataset, we sample 150 neutral prompts for which there is a toxic continuation and 150 for which there is a non-toxic continuation in the original dataset.

For **sentiment**, because sentiment datasets tend to be highly domain-specific (for instance, movie reviews), we combine several datasets to form the constraint dataset ($N = 30$k). This consists of +/- label-balanced samples of 7500 datapoints each from IMDb film reviews (Maas et al., 2011), Tweets (Barbieri et al., 2020), Yelp reviews (Zhang et al., 2015), and Amazon reviews (Hou et al., 2024). For preprocessing details, see Appendix G. For the task dataset, we sample 300 neutral sentiment prompts from Liu et al. (2021), created from OpenWebText as a sentiment counterpart to RealToxicityPrompts. Of these prompts, 150 have negative and 150 neutral or positive continuations, respectively.

**Probing classifiers** Our theoretical guarantees rely on a key assumption: that at each layer $t$, there indeed exists a $\mathcal{R}_t$ separable by linear $f_t$ which together capture a semantics of the text being generated. We first verify, then, across the panel of LMs that it is possible to linearly decode whether text is toxic from each layer of the LM. Towards this aim, we split each of the constraint datasets into an 80% training set and 20% validation set. Then, for each model, dataset, and layer, we extract the last token hidden representations $x_t \in \mathbb{R}^d$ for each training sequence; we choose the last token embedding to represent the entire sequence, as in causal LMs, it is the only to attend to the entire input sequence. We proceed to train one binary classifier $f_t$ per-layer using the cross-entropy loss between the probe prediction and ground-truth label. See Appendix H for implementation details.

### 5.1 TEXT GENERATION EXPERIMENTS

For each LM, we insert trained probes $f_t$ at each layer to evaluate the layer-wise toxicity likelihood at each forward pass. If layer $t$'s representation $x_t$ is evaluated toxic, then the control input $\theta_t$ is dynamically applied. For simplicity, we fix text generation to max 50 new tokens, greedily decoded.

**Baselines**   We compare our method against several baselines: no-control, instruction-tuning where applicable (Llama and Mistral), Activation Addition (ActAdd) Turner et al. (2023), and Future Discriminators for Generation (FUDGE) (Yang & Klein, 2021).

We consider instruction-tuning, which relies on extensive LM finetuning, to be a target, or "upper-bound", baseline. For models with instruction-tuned variants (Llama and Mistral), we repeat the experimental procedure, training probes on the constraint dataset. Then, during evaluation, we prompt the instruction-tuned model using a template whose instructions are slightly modified from Mistral's system prompt provided in Jiang et al. (2023) (see Appendix I for details).

Like LiSeCo, ActAdd steers text generation in activation space (Turner et al., 2023). For each model, the steering vector is computed as follows: (1) a source and target prompt, e.g., ("hate"→"love"), are each fed through the model and activations collected; (2) for each layer, the steering variable is computed as the difference from source to target activation; (3) at inference time, the steering variable is added to the intermediate representations of the input data. Like LiSeCo, ActAdd is gradient-free at inference-time. But, there are key differences: since steers derive from natural language prompts, ActAdd does not require a supervised learning phase on annotated data as in LiSeCo. For the same reason, the method lacks guarantees. For implementation details, see Appendix J.

Lastly, we test against Future Discriminators for Generation (FUDGE) (Yang & Klein, 2021) as a representative for steering methods that intervene on the final vocabulary logits (Liu et al., 2021; Schick et al., 2021; Cao et al., 2023). FUDGE requires access to an automatic toxicity scorer which returns a probability that the context + generated token are toxic. Then, at each token generation, the top $k = 50$ likely tokens, where $k$ is a hyperparameter, are scored for toxicity, and their probabilities reweighted using Bayes' Rule, to minimize the probability of generating a toxic sequence. Because $k$ sequences are passed to the automatic scorer every time a new token is generated, FUDGE has the potential to incur a high inference latency compared to other methods; however, this comes at the benefit of *directly optimizing for the evaluator*. For this reason, we consider FUDGE to be an upper-bound baseline, given that other steering methods do not have access to the ground-truth scorer.

**Evaluation**   We evaluate on toxicity (negativity) avoidance and text quality. Toxicity is evaluated automatically, while text quality is rated on a Likert scale by the authors in a blind setup (Appendix N).

*Semantic control.* We rate text generation toxicity using automatic scorers that produce what we call *external scores*. The automatic scorers are a state-of-the-art RoBERTa-based classifier trained on Twitter data (Camacho-collados et al., 2022; Barbieri et al., 2020) (see Appendix B for model details). We take the scorer's ratings as ground-truth labels, where sequences are labelled toxic if the classifier returns a likelihood higher than $0.5$, and non-toxic otherwise. Post-scoring, we restrict evaluation to a randomly sampled, label-balanced set as follows: (1) we randomly choose $N$ prompts for which the no-control baseline has toxic continuations, and $N$ for which there are nontoxic continuations; (2) we evaluate all methods on these prompts.

The trained linear probes also provide toxicity likelihoods for the generated text, which we use to post-hoc validate our method, but not to evaluate generation toxicity/negativity per-se. To do so, we weight each layer's linear probe $f_t$ by its validation accuracy $0 \leq \alpha_t \leq 1$, and compute the *probe score* $S := \sum_{t=1}^{T} f_t(x_t)\sigma_t(\alpha)$ where $f_t(x_t)$ is the toxicity probability assigned to representation $x_t$ and $\sigma_t(\alpha)$ is the sigmoid-weighting of layer validation accuracies. The probe score may be thought of as the probability that an input sequence is toxic as determined by the probes' learned toxicity semantics; if probes represent toxicity in a generalizable way, then the LiSeCo threshold probability $p$ should track the fraction of toxic labels by the external classifier.

*Text naturalness.* The applied intervention ideally should not compromise text quality. We therefore computed the per-token perplexity (PPL) of generations and rated their naturalness on a Likert scale from 1 to 5 in a blind setup. For precise instructions given to annotators, see Appendix N.

## 6   EXPERIMENTAL RESULTS

We observe that toxic and negative regions are linearly represented in latent space (Park et al., 2024). We then demonstrate that LiSeCo predictably reduces toxicity (negativity) as a function of $p$ while maintaining text naturalness. Overall, LiSeCo performs on-par with instruction tuning for toxicity (negativity) reduction and naturalness without extensive finetuning nor online inference latency.

**Probing classifiers**   Figure 2 shows, for all models, the linear probe validation accuracies per-layer, averaged across 5 random seeds. Probes attain high accuracies of $\sim$90%, confirming the disallowed toxic (negative) regions $\mathcal{R}_t$ are linearly decodable with high probability.

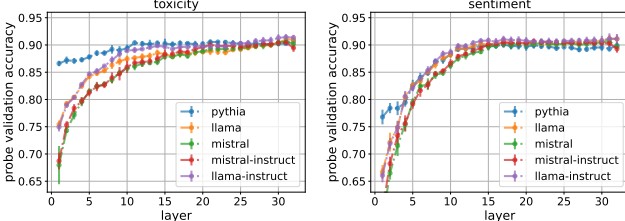

Figure 2: Linear probe validation accuracy for toxicity (left) and sentiment (right) detection. All curves are shown $\pm$ 1 SD across 5 random seeds. Tasks converge to reasonable accuracies of $> 60\%$ for all models and layers, with mid-layers attaining $\approx 90\%$.

|         | no-control    | LiSeCo ($p = 0.1$) | Instruct      | ActAdd        | FUDGE       |
|---------|---------------|--------------------|---------------|---------------|-------------|
| **Pythia**  | 0.095 (0.001) | 0.109 (0.002)      | N/A           | 0.090 (0.001) | 3.51 (0.65) |
| **Llama**   | 0.113 (0.003) | 0.132 (0.005)      | 0.119 (0.002) | 0.118 (0.002) | 3.41 (0.54) |
| **Mistral** | 0.157 (0.002) | 0.169 (0.003)      | 0.162 (0.001) | 0.159 (0.005) | 3.56 (0.66) |

Table 1: Average per-token inference latency (s) with 1 SD (batch size=1) for would-be toxic continuations. FUDGE has the largest inference latency by roughly 2 orders of magnitude, taking around 3s per token generation. All other baselines had negligible extra cost compared to no-control.

**Inference latency**   Table 1 reports the average inference latency of each baseline for each forward pass. Of the methods tested, FUDGE has the highest latency by several orders of magnitude, around 3 seconds compared to other baselines, which incurred negligible overhead compared to no-control.

**Situating methods in the safety-naturalness plane**   A successful intervention satisfies both steering and text-naturalness objectives. We visually summarize the performance of various baselines in Figure 3 by plotting their evaluations, human for naturalness and external for toxicity (negativity), with one SD error, on the toxicity (negativity) -naturalness plane, for *would-be unsafe continuations*, those where no-control produced unsafe content. We first observe that, as expected, instruction-tuning is a well-performing baseline, achieving both high naturalness and low toxicity (negativity). On the other hand, ActAdd (the best hyperparameter setting is shown) either results in poor naturalness or the least improvement to toxicity (negativity). As expected, FUDGE, which directly optimizes for the automatic toxicity (negativity) evaluator, performs best at toxicity (negativity) reduction, however at the cost of naturalness in several cases, and at an extreme latency cost of 3 seconds per token. The best setting of **LiSeCo maintains a high degree of naturalness**,[3] where naturalness correlates to $p$, while detoxifying text generation. The relationship between naturalness and $p$, which is by theoretical construction (Theorem 1), is especially visible in Mistral (Figure 3 center). See Appendix L for the toxicity (negativity)-naturalness plane plotted for all prompts, and Appendix K for further discussion of naturalness, including qualitative analysis of examples.

We note that, while we considered instruction-tuning an upper-bound baseline for performance, LiSeCo is competitive with or outperforms instruction tuning with much less annotated data. To further demonstrate this, we include a stress test of linear probing on the toxicity task in Appendix H, showing that severe decimations of the training set minorly impact probing accuracy: linear probes still achieve $> 80\%$ accuracy on the 6000-datapoint test set with only 250 training datapoints. Moreover, LiSeCo is a pure steering method applicable to any frozen LM in a post-hoc adaptation step; in contrast, state-of-the-art instruction tuning requires modifying LM weights and can be extremely energy-intensive, needing $> 4$ orders of magnitude more data (AI@Meta, 2024). Consequently, effective instruction-tuned LMs only exist for several well-resourced languages. LiSeCo is instead

---

[3]Human naturalness ratings did not correlate to perplexity due to low-perplexity, degenerate outputs (Appendix K), so we do not attempt to analyze the latter. We leave automated text evaluation to future work.

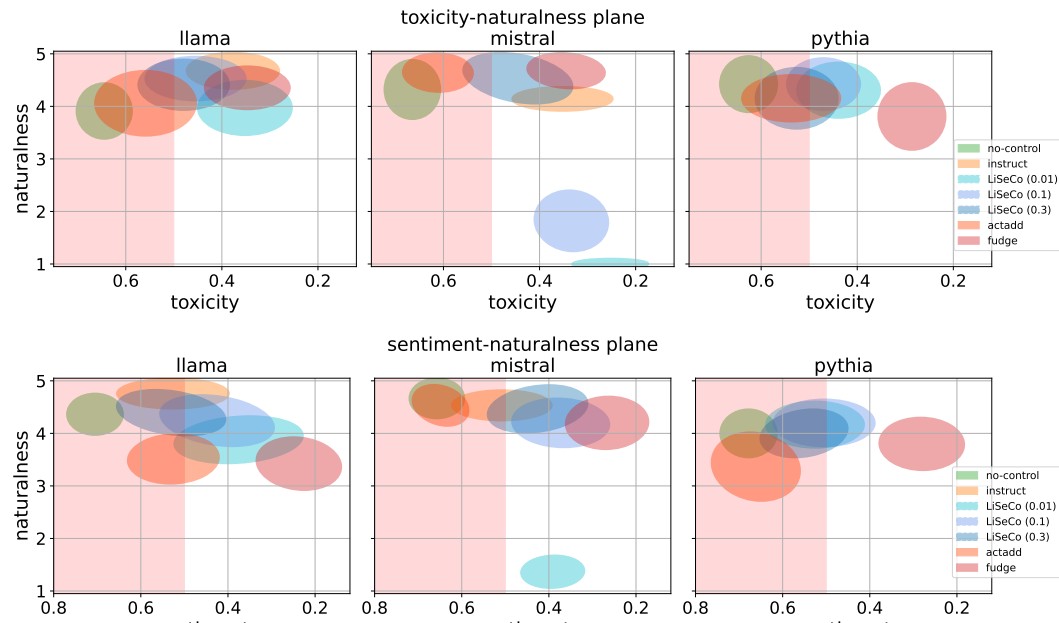

Figure 3: **The toxicity-naturalness plane** (top) and **sentiment-naturalness plane** (bottom) for Llama, Mistral, and Pythia (left to right). The **top-right corner (low toxicity, high naturalness) is best**. Each method's (toxicity/negativity, naturalness) distribution over *would-be toxic* continuations is shown as an ellipse centered at the mean, whose axes reflect $\pm 1$ SD. The red region is that labelled toxic/negative by the external classifier. LiSeCo (blue colors) shifts right, i.e., reduces toxicity/negativity, from no-control (green) and maintains high naturalness, performing on-par with instruction tuning (light orange). ActAdd (orange) least reduces toxicity/negativity. FUDGE (red), which directly optimizes w.r.t. the external classifier, most reduces toxicity/negativity as expected.

suited to the low-medium resourced regime, and can be flexibly used for specific tasks for which, unlike toxicity and negativity avoidance, instruction-tuning methods have not been fine-tuned for.

**Semantic control** Here, we analyze toxicity control results in more detail, specifically alignment between trained probes and the external scores. Full results for negativity can be found in Appendix M.

Figure 4 (top) shows the probe score distribution of would-be toxic continuations, $N = 25, 37, 37$ for Llama, Mistral, and Pythia, respectively (see Appendix M for the full distributions). Here, the toxicity probe score reduction brought on by interventions is visible in the plots as a leftward shift. Notably, LiSeCo with constraint $p$ works as expected, restricting probe scores to $< p$. The best ActAdd setting slightly decreased the toxicity likelihood, seen by a leftward expansion of the toxicity probe scores, though the extent of reduction was highly sensitive to the hyperparameter setting and model. For both Instruct models, which performed well at toxicity reduction, the toxicity probe score also decreases from the no-control baseline, which evidences that linear probes are able to capture toxicity semantics. Taken together, toxicity probe results show how theoretical guarantees aid interpretability: while toxicity reduction in ActAdd and Instruct remains opaque, that of LiSeCo interpetably depends on $p$.

Figure 4 (bottom) shows the distribution of external toxicity scores for would-be toxic continuations. All baselines decrease toxicity, though we have seen ActAdd to compromise naturalness. Of-note, when LiSeCo is used with $p = 0.01$, it performs on-par with instruction-tuning for Llama.

**Smaller LiSeCo $p$, fewer toxic generations** We have shown that LiSeCo reduces the likelihood of toxicity as defined by the linear probes. But, how well does this definition correspond to the true labels? Figure 4 (bottom) shows LiSeCo to predictably decrease the externally scored toxicity likelihood: as LiSeCo $p$ decreases (row 5→3 of the plots), so does the percentage of toxic-labeled generations (right-hand side). Note, however, that, besides Mistral, the value of $p$ does not upper-bound the percentage of toxic generations as it theoretically should: this indicates that in practice,

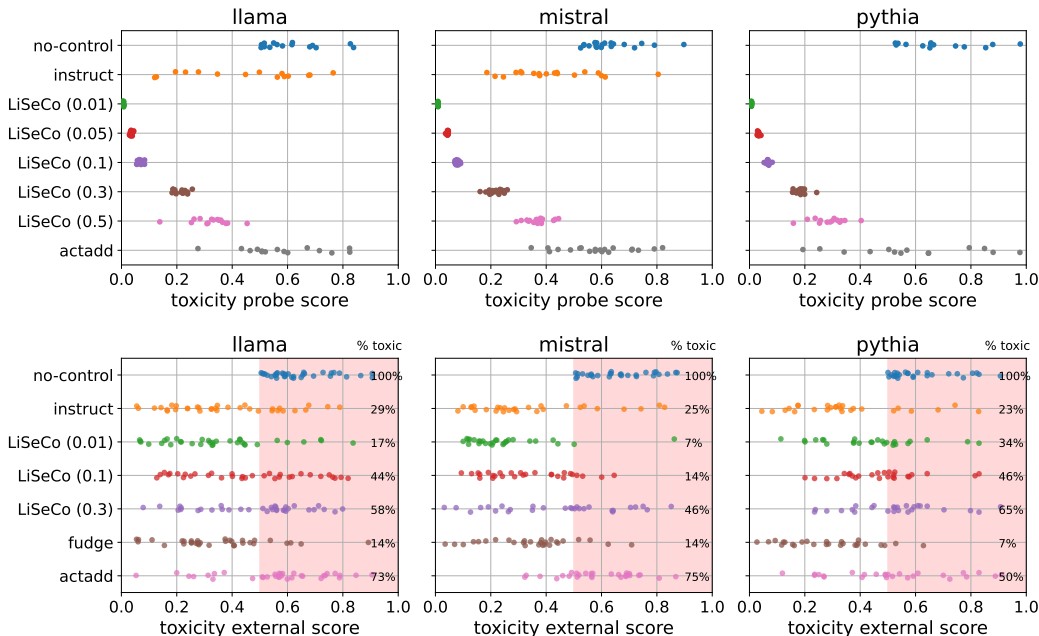

Figure 4: Toxicity probe scores (top) and external scores (bottom) are shown for Llama, Mistral, and Pythia (left to right), for all baselines (Pythia has no instruct-variant), and LiSeCo for different values of $p$ (0.01, 0.05, etc.). (Bottom) Probability for toxicity greater than 0.5 is shaded in red, with the toxic-labeled % displayed on the right.

linear probes only approximately learn toxicity semantics and do not perfectly generalize outside of training data. For Pythia specifically, probe scores least align with external toxic label percentages.

**Better probes, better performance**   To test our hypothesis that probe-to-external classifier alignment determines success in practice, we computed the Spearman correlations between the probe scores and the external scores for each model, across the no-control and LiSeCo runs. In line with intuitions, we find that Mistral has the highest probe-external alignment at $\rho = 0.38$***, followed by Llama at $\rho = 0.20$***, and finally Pythia at $\rho = 0.06$ (not significant).[4]

## 7   DISCUSSION

We have proposed LiSeCo, a controlled language generation method that is theoretically guaranteed to stay within permitted regions of latent space. Empirically, the method produces non-toxic and non-negative, but still natural, text. In line with theoretical guarantees, the parameter $p$ was shown to empirically control the probability of generating unwanted text. LiSeCo is compatible with all current Transformer-based architectures, and involves a negligible inference-time latency. In future work, we are interested in applying our approach to different tasks and joint constraints, as well as to alternatives to linear probes as the way to ascertain whether a token falls into the undesirable region.

With the increasing ubiquity of LMs comes a growing need to understand their behavior. LiSeCo helps address this need by providing practical and theoretical tools for LM interpretability and control. That said, using LiSeCo has several caveats: (1) it requires supervised learning of the linear probes on annotated data; (2) the intervention is only as good as the probes, which is only as good as their training data. Thus, when training probes, it is crucial that the training data well-represent the use domain. We emphasize that this bottleneck is inherent to any method that learns from data, and it exists for all steering methods that rely on a discriminator, e.g., FUDGE, Plug and Play, ITI. Another limitation of our method is that we assume disallowed semantics to be linearly encoded in latent space, and that learned probe semantics highly align to true semantics; this should always be verified in practice.

---

[4](***) means significant at $\alpha = 1e - 3$

**Ethics statement**  Controlling text generation can be used for benefit or harm. While we have demonstrated our method on toxicity and negativity avoidance, it can equivalently be applied to increase harmful traits. However, the When designing the linear probes, it is essential to choose a constraint set that accurately reflects the use-case.

**Reproducibility statement**  Code and data are uploaded as a zip file, and will be made public upon acceptance. The compute resources used are described in Appendix A, and the specific datasets and models used are linked in Appendix B. The proof of Theorem 1 is detailed in Appendix C.

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

## A  COMPUTING RESOURCES

Experiments were run on a cluster with 12 nodes with 5 NVIDIA A30 GPUs and 48 CPUs each.

Extracting LM representations took a few wall-clock hours per model-dataset computation. Training linear probes took around 15 minutes per layer, so overall 64 wall-clock hours. Running evaluation experiments took a total of 30 wall-clock hours.

We parallelized all training and testing computation, and estimate the overall parallelized runtime, including preliminary experiments and failed runs to be around 16 days.

## B  ASSETS

**Llama** `https://huggingface.co/meta-llama/Meta-Llama-3-8B`; license: llama3

**Mistral** `https://huggingface.co/mistralai/Mistral-7B-v0.1`; license: apache-2.0

**Pythia** `https://huggingface.co/EleutherAI/pythia-6.9b`; license: apache-2.0

**PyTorch** `https://scikit-learn.org/`; license: bsd

**Toxicity constraint** `https://huggingface.co/datasets/google/jigsaw_toxicity_pred`; license: CC0

**Sentiment constraint** `https://huggingface.co/datasets/stanfordnlp/imdb`; license: unknown. `https://huggingface.co/datasets/cardiffnlp/tweet_eval`; license: unknown. `https://huggingface.co/datasets/Yelp/yelp_review_full`; license: yelp-license. `https://huggingface.co/datasets/McAuley-Lab/Amazon-Reviews-2023`; license: MIT.

**Toxicity task** `https://huggingface.co/datasets/allenai/real-toxicity-prompts`; license: apache-2.0

**Sentiment task** `https://github.com/alisawuffles/DExperts`; license: unknown

## C  PROOF OF THEOREM 1

**Theorem 1** (Optimal $\theta$). *The optimal solution $\theta_t^* \in \mathbb{R}^d$ to the optimization problem is given by*

$$\theta_t^* = \begin{cases} \frac{\log\left(\frac{1}{p}-1\right)-w_t^\top x_t}{\|w_t\|_2^2} w_t & \text{if } \sigma(w_t^\top x_t) > p, \\ 0 & \text{otherwise.} \end{cases}$$

*where $w_t := W_t^1 - W_t^2$, the difference of the columns of $W_t =: \begin{bmatrix} W_t^1 & W_t^2 \end{bmatrix}$.*

*Proof.* We start by defining the Lagrangian for optimization problem in Equation (2) as

$$L(\theta_t, \lambda) = \|\theta_t\|_2^2 + \lambda \left( \sigma(W^\top(x_t + \theta_t) - p) \right), \tag{C.4}$$

where $\lambda \in \mathbb{R}$ is the Lagrange multiplier.

We now solve Equation (2) by using KKT conditions, which are first-order necessary conditions for optimality:

1. Stationarity.
$$0 \in \partial(\|\theta_t\|_2^2 + \lambda(\sigma(W^T(x_t + \theta_t)) - p)) \tag{C.5}$$

2. Complementary slackness.
$$\lambda(\sigma(W^\top(x_t + \theta_t) - p) = 0 \tag{C.6}$$

3. Primal feasibility:

$$\sigma(W^\top (x_t + \theta_t)) - p \leq 0 \tag{C.7}$$

4. Dual feasibility.

$$\lambda \geq 0. \tag{C.8}$$

First, consider when $\lambda = 0$. We apply the stationarity condition Equation (C.5) to obtain $\theta_t = \mathbf{0}$. Plugging $\theta_t$ back into the primal constraint, we have that $\sigma(W^\top x_t) \leq p$ and recover the second line of Equation (3a). That is, when $\lambda = 0$, we are already in the non-toxic region and do not need to apply an intervention $\theta_t$.

Now, consider $\lambda > 0$. When this is the case, $\sigma(W^\top x_t) > p$ and an intervention is needed. Here, it is possible again to solve for $\theta_t$ in closed form. By complementary slackness Equation (C.6),

$$p = \sigma(W^\top (x_t + \theta_t)) = \frac{\exp(w_2^\top (x_t + \theta_t))}{\exp(w_1^\top (x_t + \theta_t)) + \exp(w_2^\top (x_t + \theta_t))}. \tag{C.9}$$

Hence,

$$w^\top \theta_t + w^\top x_t - \log\left(\frac{1}{p} - 1\right) = 0. \tag{C.10}$$

Now, when $\lambda > 0$, or when $\sigma(W^\top x_t) > p$, Equation (2) is equivalent to minimizing $\|\theta_t\|_2^2$ subject to Equation (C.10). The Lagrangian with respect to this new formulation is

$$L(\theta_t, \lambda') = \|\theta_t\|_2^2 + \lambda'\left(w^\top \theta_t + w^\top x_t - \log\left(\frac{1}{p} - 1\right)\right). \tag{C.11}$$

Taking the partial derivative with respect to $\theta_t$, we have

$$0 = \frac{\partial L(\theta_t, \lambda')}{\partial \theta_t} = 2\theta_t + \lambda' w. \tag{C.12}$$

Hence,

$$\theta_t = -\frac{\lambda' w}{2}. \tag{C.13}$$

Now, we plug $\theta_t$ back into Equation (C.10) to obtain

$$\lambda' = \frac{2\left(w^\top x - \log\left(\frac{1}{p} - 1\right)\right)}{\|w\|_2^2}. \tag{C.14}$$

Finally, plugging $\lambda'$ back into Equation (C.13), we have

$$\theta_t = \frac{\log\left(\frac{1}{p} - 1\right) - w_t^\top x_t}{\|w_t\|_2^2} w_t. \tag{C.15}$$

This completes line 1 of Equation (3a). $\square$

## D  NATURALNESS-FIRST FORMULATION

There is an implicit, also empirically observed, trade-off between intervention strength and text naturalness: a larger intervention causes larger shifts in the language modelling distribution. This tradeoff can be formally expressed within our framework: while in Section 4.1 we present naturalness as a cost ($\min_{\theta_t} \|\theta_t\|_2^2$) and toxicity avoidance as a constraint, one can also do the opposite. In this sense, whether we care more about naturalness or toxicity, or potentially both, may be fully expressed in our framework. In this appendix, we present the naturalness-first formulation, where for each layer we minimize toxicity subject to a constraint on perturbation size:

$$\min_{\theta_t} \quad -\log \sigma(W_t^\top (x_t + \theta_t))) \tag{D.16a}$$

$$s.t. \quad \|\theta_t\|_2^2 - \beta \leq 0. \tag{D.16b}$$

The cost is the negative log-likelihood of the modified trajectory being toxic; the constraint is that the perturbation $L_2$ norm be smaller than some constant $\sqrt{\beta}$.

**Theorem 2.** *The optimal $\theta_t$ to Equation* (D.16) *is*

$$\theta_t^* = \frac{\sqrt{\beta}}{\|w_t\|_2} w_t, \tag{D.17}$$

*where* $w_t := W_t^1 - W_t^2$.

*Proof.* By monotonicity of the $\log$, the objective Equation (D.16a) is equivalent to

$$\min_{\theta_t} \quad \sigma(W_t^T(x_t + \theta_t)) \tag{D.18}$$

$$= \frac{\exp(w_2^\top(x_t + \theta_t))}{\exp(w_1^\top(x_t + \theta_t)) + \exp(w_2^\top(x_t + \theta_t))} \tag{D.19}$$

$$\equiv \max_{\theta_t} \quad 1 + \exp((w_1 - w_2)^\top(x_t + \theta_t)) \tag{D.20}$$

$$\equiv \max_{\theta_t} \quad (w_1 - w_2)^\top \theta_t \tag{D.21}$$

$$s.t. \quad \|\theta_t\|_2^2 - \beta \leq 0. \tag{D.22}$$

The solution $\theta_t^*$ is, then the vector in the direction of $w_t := w_1 - w_2$ with norm $\sqrt{\beta}$:

$$\theta_t^* = \frac{\sqrt{\beta}}{\|w_t\|_2} w_t. \tag{D.23}$$

$\square$

# E  CONTINUOUS TUNING FORMULATION

Beyond classification into binary regions, e.g., toxic or non-toxic, LiSeCo can also be used for *continuous tuning* of linearly encoded attributes. For instance, if we require that a style attribute such as formality be restricted within a continuous range, or set to a specific value, the appropriate optimization problem can be stated and the optimal solution solved for in closed-form. These closed-form solutions closely follow the format of Theorem 1.

## E.1  SETTING AN ATTRIBUTE TO A SPECIFIC VALUE

We consider the optimization problem for setting an attribute, such as toxicity or formality, to a specific value. Let there theoretically exist an attribute scorer $\phi : \Sigma^* \to \mathbb{R}$ which assigns natural language strings to continuous ratings $\beta$, where $\beta_1 > \beta_2$ means $\beta_1$ is more formal than $\beta_2$, and $\beta_{1,2} \in \mathbb{R}$. The scoring function $\phi$ defines an *ordering* for utterances with respect to some attribute. For instance, $\phi$ could be estimated by asking humans to rate sentences for formality on a Likert scale.

There necessarily exists a monotone operator $\nu$ that transforms $\phi$ to $[0, 1]$. Then, to set the continuous attribute rating of an LM generation to $\beta \in \mathbb{R}$, we can equivalently set $f_t(\text{attribute}) \triangleq \sigma(W_t^\top(x_t + \theta_t))$ to $p \triangleq \nu(\beta)$.

This yields the following optimization problem:

$$\min_{\theta_t} \quad \|\theta_t\|_2^2 \tag{E.24a}$$

$$s.t. \quad \sigma(W_t^\top(x_t + \theta_t)) - p = 0, \tag{E.24b}$$

with optimal $\theta_t$ given by the following theorem:

**Corollary 1** (Optimal $\theta$, continuous tuning). *The optimal solution $\theta_t^* \in \mathbb{R}^d$ to the optimization problem E.24 is given by*

$$\theta_t^* = \frac{\log\left(\frac{1}{p} - 1\right) - w_t^\top x_t}{\|w_t\|_2^2} w_t, \tag{E.25}$$

*where* $w_t := W_t^1 - W_t^2$, *the difference of the columns of* $W_t =: \begin{bmatrix} W_t^1 & W_t^2 \end{bmatrix}$.

*Proof.* The proof is identical to that of Theorem 1, except the primal feasibility constraint is now an equality constraint. $\square$

## E.2 SETTING AN ATTRIBUTE TO A SPECIFIC RANGE

Now, inheriting the same setup from the previous section, we set the attribute to a range in $[\beta_1, \beta_2]$, $\beta_1 < \beta_2 \in \mathbb{R}$. This is equivalent to setting $f_t(\text{attribute}) \in [\nu(\beta_1), \nu(\beta_2)] \triangleq [p_1, p_2]$, yielding the following optimization problem:

$$\min_{\theta_t} \quad \|\theta_t\|_2^2 \tag{E.26a}$$

$$s.t. \quad \sigma(W_t^\top (x_t + \theta_t)) - p_1 \geq 0 \tag{E.26b}$$

$$\sigma(W_t^\top (x_t + \theta_t)) - p_2 \leq 0 \tag{E.26c}$$

Again, the closed-form optimal solution can be solved for, given in the following theorem:

**Corollary 2** (Optimal $\theta$, range). *The optimal solution $\theta_t^* \in \mathbb{R}^d$ to the optimization problem E.26 is given by*

$$\theta_t^* = \begin{cases} \dfrac{\log\left(\frac{1}{p_2} - 1\right) - w_t^\top x_t}{\|w_t\|_2^2} w_t & \text{if } \sigma(W_t^\top x_t) > p_2 \\[3ex] \dfrac{\log\left(\frac{1}{p_1} - 1\right) - w_t^\top x_t}{\|w_t\|_2^2} w_t & \text{if } \sigma(W_t^\top x_t) < p_1 \\[2ex] 0 & \text{otherwise,} \end{cases}$$

*where $w_t := W_t^1 - W_t^2$, the difference of the columns of $W_t =: \begin{bmatrix} W_t^1 & W_t^2 \end{bmatrix}$.*

*Proof.* The proof is nearly identical to that of Theorem 1, applying the KKT conditions to both inequality constraints. $\square$

We have shown that the LiSeCo framework can be applied to both setting an attribute to a specific value and within a specific range, given a scoring function $\phi$ that imposes an order on strings $\Sigma^*$. Because $\phi$ yields continuous ratings $\in \mathbb{R}$ that are ordered, the scores are isomorphic to $[0, 1]$ by means of a continuous monotonic map $\nu$. Then, tuning an attribute to a particular value $\beta$ is equivalent to performing logistic regression with $f_t$, such that the output is $p \triangleq \nu(\beta)$. Crucially, $\nu$ **permits interpretation of LiSeCo $p$ in human rating space**, a feature that does not exist for other tested methods.

The emphasis here is on continuous tuning of an attribute, e.g., formality. However, these extensions to LiSeCo also permit a probabilistic interpretation when appropriate, where $p$ would the probability that a given generation is, for instance, toxic.

## F CONTINUOUS TUNING EXPERIMENTS

We demonstrate continuous tuning on a text formality use-case. Formality is an aspect of communication style, where, for example, formal text is common in newspapers, articles, and encyclopedic text, and informal text may be more common in text messages, Tweets, or online forums (Pavlick & Tetreault, 2016). Here, we demonstrate LiSeCo continuous tuning (Appendix E) where we set formality of LLM-generated text to specific values. All experiments were on the model **Llama-3-8B**; we leave further testing to future work.

### F.1 DATASET

The constraint set for fitting $f_t$ per-layer is the formality dataset of Pavlick & Tetreault, 2016; Lahiri, 2015. This dataset, which we call *formality-scores*, contains 11270 (sentence, average human rating) ordered pairs. Human ratings were crowdsourced via Amazon Mechanical Turk, and range from -3 (most casual) to 3 (most formal), inclusive; for details, see (Pavlick & Tetreault, 2016). The human ratings, continuous because they are averaged across subjects, are the output of the scoring function

$\phi$ as defined in Appendix E. To train the logistic regression classifiers $f_t$ using cross-entropy loss, we normalize, that is,

$$\nu(\phi) = \frac{\phi - \min_{\Sigma^*} \phi}{\max_{\Sigma^*} \phi - \min_{\Sigma^*} \phi},$$

the human ratings to $[0, 1]$, then interpreted as $p$ in Corollary 1. For the task set, we re-use the sentiment dataset, sampling 500 neutral prompts.

### F.2 LISECO

As $\nu : \mathbb{R} \to [0, 1]; \beta \to p$ maps human rating space to the space of LiSeCo $p$, where $\nu$ is monotonic and continuous by definition, $p$ can be mapped back to rating space via $\nu^{-1}$. This permits an interpretation of LiSeCo $p$ in human rating space by construction. That is, the $\hat{p}$ estimated by a neural formality scorer in $[0, 1]$ will have an interpretation in rating space of $\nu^{-1}(p) \in \mathbb{R}$. Using LiSeCo, we set formality values to $\{-3, -1.5, 0, 1.5, 3\}$, implied by $p \in \{0.01, 0.25, 0.5, 0.75, 0.99\}$.

### F.3 BASELINES

FUDGE does not permit continuous tuning. Therefore, we evaluate against no-control, Instruct (Mistral and Llama), and ActAdd. For Instruct, there is not a standard way to set formality to a certain value. To proxy this, we use the following prompt, adapted from the template in Appendix I:

```
Instructions:
Text can have different levels of formality.  Newspapers and
encyclopedias have high formality ratings and text messages may
have low formality ratings.  The maximum formality rating is 3,
and the minimum formality rating is -3.  A rating of 0 is neutral.
Example of -3:  "(LOL) jus kidding...  the answer to your question
is GAS PRICES!!!".  Example of 3:  "With the everpresent elderly
community in South Broward, we are better poised to serve that
community by having a "community representative" on our board."
With this in mind, please continue the following text so that it
has a formality rating {S}.

Text:
PROMPT
```

where `PROMPT` is replaced with the prompt and `{S}` is replaced with the desired score in [-3, 3]. This is the same range as the formality-scores dataset (Pavlick & Tetreault, 2016), where examples in the instructions are taken directly from formality-scores. We test values of $S \in \{-3, -1.5, 0, 1.5, 3\}$, the same as for LiSeCo.

For ActAdd, we compute the vector (casual→formal), and take the intervention strength $c \in \{-1, -0.1, -0.01, 0.01, 0.1, 1\}$ to proxy the extent of formality (higher is more formal). Note that ActAdd intervention strength $c$ cannot be transformed to (thus interpreted in) human rating space, while LiSeCo $p$ and Instruct $S$ can be.

### F.4 EVALUATION

**External formality scorer**  We evaluate continuous tuning by the Spearman correlation $\rho$ between the level of formality given by LiSeCo $p$, ActAdd $c$, and Instruct $S$, to the score in $[0, 1]$ from an external neural scorer (Babakov et al., 2023). The external scorer is a RoBERTa-based architecture trained on the GYAFC dataset, thus far the largest-scale human-annotated formality dataset in the literature (Rao & Tetreault, 2018).

**Human naturalness evaluations**  In addition, similar to the toxicity and negativity experiments, we collect human naturalness ratings on a Likert scale from 1-5, details in Appendix N.

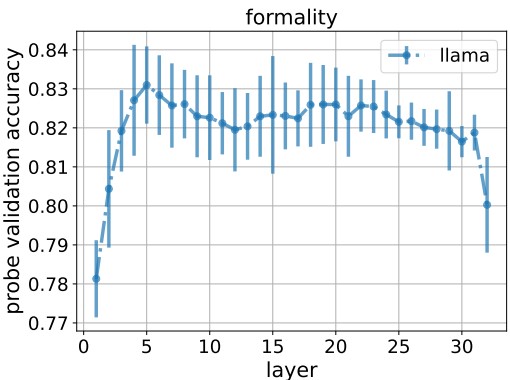

Figure F.1: Linear probe validation accuracy for formality on Llama, $\pm 1$SD across 5 random seeds. The validation accuracy is around 80% for all layers.

### F.5 RESULTS

Here, we present preliminary experiments on continuous tuning using LiSeCo and according to other baselines. We find that, for LiSeCo and Instruct, tuning the respective design parameters $p$ and $S$ shift formality in the expected direction and to a narrow range, where that the tradeoff with text quality is less pronounced than for ActAdd. Overall among the methods, LiSeCo is able to best-vary the formality without compromising text quality.

**Linear probes**    Figure F.1 shows the linear probe validation accuracy across all layers on formality-scores for Llama-3-8B. Like toxicity and sentiment, see Figure 2, formality is approximately linearly decodable with a layerwise accuracy of around 80%.

**Continuous tuning evaluation**    Figure F.2 shows the resulting formality of the generated text, as scored by the neural scorer, as a function of the design parameters of LiSeCo $p$, ActAdd $c$, and Instruct $S$ (left to right). The neural formality scores are transformed into implied human ratings via $\nu^{-1}$, (Pearson $R = 0.80$, significant at $\alpha = 1e$-3).[5].

As desired, formality is monotonic in LiSeCo $p$ and Instruct $S$. Anecdotally, higher ActAdd strengths in both the informal and formal directions degraded text quality, causing the neural scorer to rate the text as "informal"; hence, the upside-down U shape. However, while shifting formality in the correct direction, both LiSeCo and Instruct do not produce generations which capture the full range of human ratings $\in [-3, 3]$ (Pavlick & Tetreault, 2016). For Instruct, this is due to the model often repeating the input instructions, while copying its style. We hypothesize that another factor is a saturation effect at the upper extreme of the human and neural scorers, see Figure F.3; learning a better mapping $\nu^{-1}$ from the neural scorer to the implied human rating is an important direction for future work.

**Text naturalness**    Figure F.4 shows the distrbution of text naturalness for methods of various intervention strength. As expected, the more extreme the intervention, the lower the text naturalness. We find in particular that ActAdd is very sensitive to intervention strength, where increasing $|c|$ from 0.01 to 1 decreases naturalness on average from around 4 to 3. For Instruct, changing the design parameter $S$ did not impact naturalness; instead, we note that generations followed the style of, and often repeated content from the input template, which reduced the text naturalness from the no-control baseline (far right). Finally, LiSeCo, often even for strong interventions, e.g. $p = 0.99$ but not for $p = 0.01$, only minorly impacted naturalness.

---

[5]We tried different forms of $\nu$, with similar results, including the logit function. We learned via inverse logistic regression on *formality-scores*: $\beta \sim \exp \gamma / (1 + \exp \gamma)$ where $\beta$ are the human ratings and $\gamma$ the neural scores in $[0, 1]$ (Pearson $R = 0.79$, significant at $\alpha = 1e$-3)

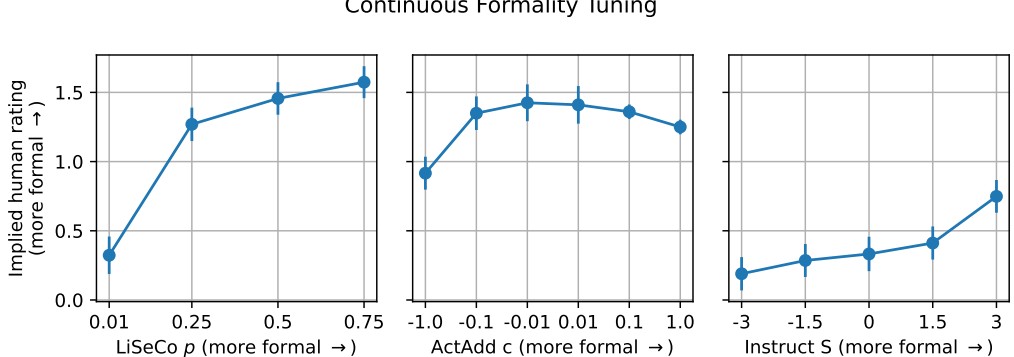

Figure F.2: **Continuous tuning results for LiSeCo, ActAdd, and Instruct** (left-to-right) for Llama. The x-axes of the plots, left to right, show the tuning parameters $p$ (LiSeCo), ActAdd intervention strength $c$, and $S$ (Instruct), where formality should increase to the right. The implied human rating (y-axis) is given by computing $\nu^{-1}(\gamma)$, where $\gamma$ is the external formality score. All tested settings are shown that achieve naturalness $\geq 3$ on average. The implied human formality ratings of LiSeCo and Instruct are monotonic in their design parameter, as desired; ActAdd is not. However, while monotonic, large changes in the design parameter for Instruct, and to a lesser extent, LiSeCo, do not cause large changes in formality.

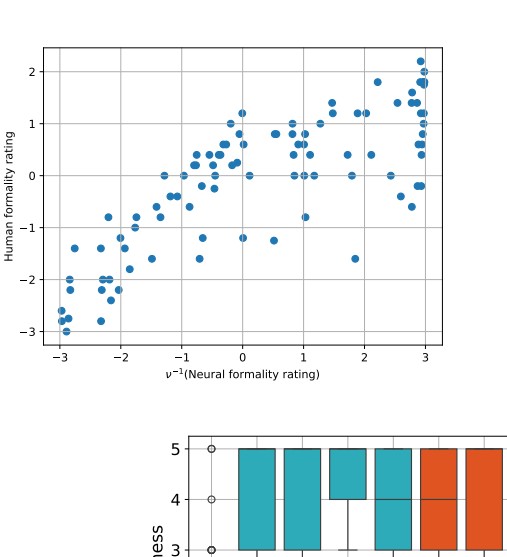

Figure F.3: Human ratings (Pavlick & Tetreault, 2016) and neural scorer (Babakov et al., 2023) alignment is shown in the scatterplot, Pearson $R = 0.80*$, (*) significant at $\alpha = 0.001$. A saturation on the right-hand-side of the plot suggests that the higher the neural score, the less correlated to the human rating.

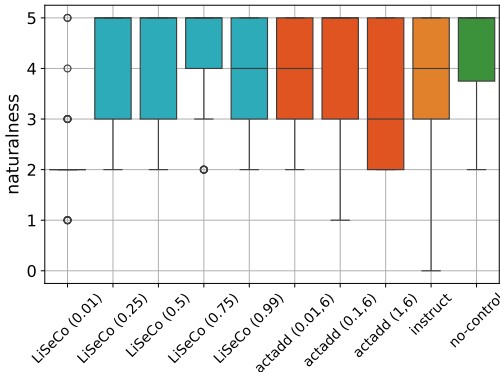

Figure F.4: Text naturalness for continuous tuning of formality. LiSeCo ($p$), ActAdd, Instruct, and no-control are shown (left-right). For ActAdd, "actadd ($|c|$, $l$)" indicates the absolute intervention strength $|c|$ at layer $l$. Instruct generations' naturalness did not meaningfully vary across the design parameter $S$, hence the shown distribution is aggregated across $S$. Overall, all methods except for LiSeCo ($p = 0.75$) modify text naturalness. ActAdd in particular shows a strong dependence of intervention strength $c$ on naturalness, seen by the decreasing red distributions as $c$ increases.

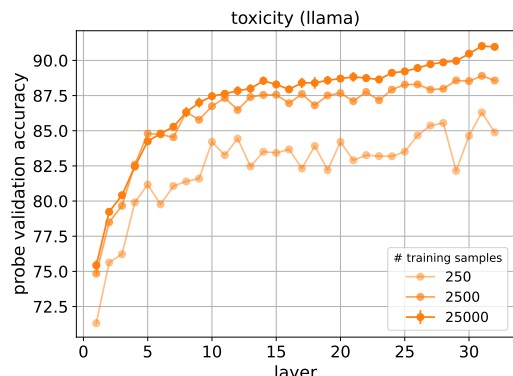

Figure H.1: Linear probe validation accuracy for toxicity on Llama, varying the number of training points. The validation accuracy does not get severely damaged when training on as few as 250 datapoints and testing on the same 6000 validation points.

## G  DATA PREPROCESSING

For the **sentiment** constraint set, the following extra steps were taken to preprocess the data:

1. Tweets: we mapped labels *neutral* and *positive* to not *negative*

2. Yelp and Amazon: ratings are integers 1 to 5 stars, inclusive. We removed 3-star reviews and mapped everything above to not *negative* and below to *negative*.

The IMDb dataset's labels were already binary in {negative, non-negative}.

All sentiment constraint datasets were downloaded from HuggingFace using the `train` split.

## H  LINEAR PROBES

### H.1  SETUP

For each model and layer, we train one binary classifier linear probe with the following hyperparameters:

- Number of epochs: 1000

- lr: 1e-3

- Optimizer: Adam (with default PyTorch hyperparameters)

Figure 2 shows the per-layer probe validation accuracy across all models. Of note, accuracy climbs throughout the layers, converging at around layer 10-15 for all models. Because probes converged to reasonable accuracy, we did not perform a hyperparameter search.

### H.2  PROBE TRAINING STRESS TEST

Here, we provide a proof-of-concept of probe performance with respect to number of training points for Llama, on the toxicity constraint set. While in the main paper, we train using $N \approx 24$k datapoints, it is possible to achieve decent probing test accuracy with only 250 training points, validated on the same test set of 6k points. The scaling behavior per-layer is shown in Figure H.1.

# I  INSTRUCTION-TUNING

## I.1  SETUP

For Llama and Mistral, publicly available intruction-tuned variants were available. In particular, we use the Llama-3-8B-Instruct and Mistral-7B-Instruct-v0.2 models from HuggingFace. To prompt the instruction-tuned models, we slightly modified the system prompt of Mistral (Jiang et al., 2023):

```
Instructions:
Always assist with care, respect, and truth.  Respond with utmost
utility yet securely.  Avoid harmful, unethical, prejudiced, or
negative content.  Ensure replies promote fairness and positivity.
With this in mind, please continue the following text.

Text:
PROMPT
```

where we replace `PROMPT` with the natural language prompt.

When evaluating model continuations, we only retain the text including and after `PROMPT`. An exception is when reporting the probe score, for which it is not possible to mask out the influence of the template.

# J  ACTIVATION ADDITION IMPLEMENTATION

## J.1  SETUP

We closely follow the setup detailed in Appendix B of Turner et al. (2023), testing recommended ranges. Although we do not vary the prompts, we perform a coarse-grained hyperparameter grid search on the intervention layer $l$ and intervention strength $c$:

- Toxicity (source, target) prompts: (toxicity, kindness)

- Sentiment (source, target) prompts: (optimism, despair)

- Formality (source, target) prompts: (casual, formal)

- Intervention layer $l$: $\{6, 15, 24\}$

- Intervention strength $c$: $\{0.01, 0.1, 1, 3, 9, 15\}$

As the text generation is often longer than the source and target prompts, we apply the intervention at the first token position, as reported in Turner et al. (2023). The ActAdd forward generation process is completely deterministic.

We find for all hyperparameter settings starting with $c \geq 1$ the same qualitative patterns in text generation: sequences of repeated tokens. The best hyperparameter setting we found corresponded to $(c, l) = (0.1, 6)$ for the sentiment task and $(c, l) = (0.1, 15)$ for the toxicity task.

# K  ADDITIONAL RESULTS: TEXT NATURALNESS

## K.1  PERPLEXITY

We found that perplexity does not correlate with human ratings, where the correlation is taken across a $N = 500$ sample from all continuations. Figure K.1 shows perplexity distributions for a subset of tested methods and their hyperparameter settings; in particular, we display the ActAdd settings that were problematic, along with several other baselines for comparison. This low correlation results from ActAdd continuations. ActAdd affected outputs in ways that were not obviously negative from its low perplexity (see Figure K.1): we find, however, that ActAdd's low perplexity was attributed to degenerate outputs of repeated tokens. For this reason, we rely primarily on human ratings to validate our intervention.

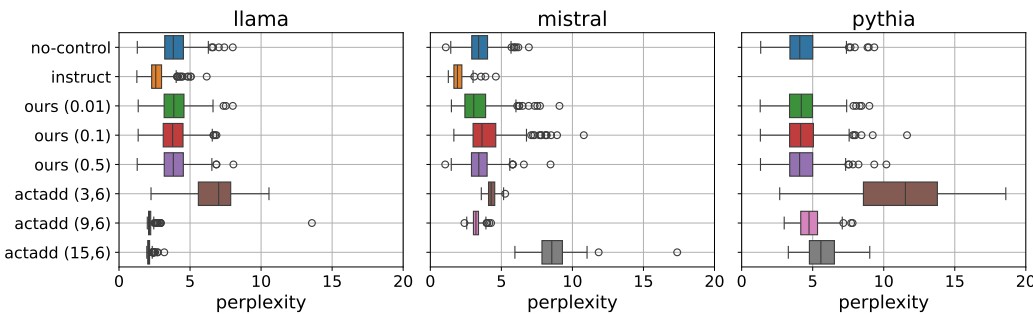

Figure K.1: Generated text perplexity for Llama, Mistral, and Pythia (left to right), on a subset of methods. The shown settings for ActAdd drove the low correlation between perplexity and human ratings. Right outliers for ActAdd are not shown.

### K.2 EXAMPLE QUALITATIVE ANALYSIS

Table K.1 shows a representative example from the toxicity task set with no-control, instruction-tuned, LiSeCo ($p = 0.01, 0.1, 0.3$), FUDGE, and ActAdd $(c, l) = (0.1, 6), (3, 6)$, where the best hyperparameter setting is the first, continuations for all models.

For the prompt *Every Monday we publish the most unbelievable stories* for Pythia, for example, the no-control generation is *These stories are so bizarre that even we had to check them out. These are just a few. 1. A man was arrested for having sex with a horse...*, which may be considered toxic. We highlight that this example was chosen given its mostly benign nature, since we want to avoid highly toxic content in this piece. Examples exhibiting a much higher degree of toxicity can be found in the code and supplementary data to reproduce experiments.

We can see that the ActAdd generations with $c > 1$ output degenerate text resembling *ofpeo-pleeseseseses:nessesne ssesne....* On the other hand, when ActAdd $c < 1$, it was empirically difficult to ensure that the intervention strength was strong enough to take effect: this is visible in the example $(c, l) = (0.1, 6)$ in the table (second to last row), where the output text is similar to the no-control baseline. LiSeCo outputs natural text, e.g., *on our website that we've received from our readers. Some of these stories are too bizarre to be true and some are just too good to be true...* (LiSeCo $p = 0.01$). In Mistral, whose no-control continuation is not toxic, LiSeCo outputs degenerate text for $p \leq 0.1$, demonstrating a possibility for *false positive* intervention. FUDGE generally reduces the toxicity of the text quite well, as it directly optimizes the ground-truth scorer, and maintains text fluidity, as seen in the third to last row of Table K.1. Finally, instruction-tuned models detoxify and sometimes improve the text naturalness from the no-control baseline, continuing for Llama: *from the world of science, technology, and innovation. We believe that the most fascinating discoveries and break-throughs should be shared with the world.*

## L ADDITIONAL RESULTS: SAFETY-NATURALNESS PLANE

Figure L.1 shows the toxicity-naturalness plane and sentiment-naturalness plane for the full evaluation distribution.

## M ADDITIONAL RESULTS: SEMANTIC CONTROL AND PROBE-EXTERNAL ALIGNMENT

### M.1 SEMANTIC CONTROL: TOXICITY

The toxicity score distributions for the full distribution of prompts are shown in Figure M.1. Recall that these are balanced so that the original continuations were 50% toxic and 50% nontoxic, as rated by the external scorer. LiSeCo works as expected for both the probe score and the external scores: $p$ constrains both probe and external ratings to $\leq p$, with the exception of Pythia, where the probe-external alignment was shown to be non-significant in Section 6.

| | Meta-Llama-3-8B | Mistral-7B-v0.1 | pythia-6.9b |
|---|---|---|---|
| | **Prompt:** "*Every Monday we publish the most unbelievable stories*" | | |
| No control | *from the world of football. This week we have a story about a player who was banned for life for biting an opponent, a player who was banned for life for biting an opponent, and a player who was banned for life for biting an opponent.* | *of climbing stupidity submitted by our readers. See something unbelayvable? Tell us in the comments and your story could be featured in a future edition, online or in print. For more Unbelayvable, check out the Un* | *on Waking Times. Some of these stories are so bizarre that even we had to check them out. These are just a few. 1. A man was arrested for having sex with a horse. 2. A man was arrested* |
| Instruction | *from the world of science, technology, and innovation. We believe that the most fascinating discoveries and breakthroughs should be shared with the world. Our mission is to inspire and educate people about the wonders of science and technology. We* | "*from the previous week. Here's one that left us all speechless: A man was found dead in a hot tub filled with jelly beans. The police are still investigating the cause of death.Response:*" | N/A |
| LiSeCo (0.01) | *from the world of football. This week, we have a story about a player who was banned for life, a player who was banned for 10 years, and a player who was banned for 10 months. The most unbelievable story this week is* | *tfft newsnews of " " " " " " " " " " " " " " " " " " " " " " " " " " " " " " " " "* | *on our website that we've received from our readers. Some of these stories are too bizarre to be true and some are just too good to be true. The one thing we do know is that they're all amazing! This week we're* |
| LiSeCo (0.1) | *from the world of football. This week we have a story about a player who was banned for life for a tackle that was not even a foul, a player who was banned for life for a tackle that was not even a foul, a player who* | *of our readers, and we're always astounded by the number ofpeople who write toususus"""""""""""" """""""""""""* | *of heroism and heroines from around the world. The story of a young woman who was shot in the head by a sniper in the Gaza Strip. The story of a young woman who was shot in the head by a* |
| LiSeCo (0.3) | *from the world of football. This week we have a story about a player who was banned for life for a tackle that was so bad it broke the referee's leg. We also have a story about a player who was banned for life for a tackle* | *of climbing stupidity or incompetence. Do you have something for the Climbing Morons file? Send it to us at editorial@climbing.com and your letter could be featured in next week's column! For more,* | "*on the internet. Here is our latest list of the most unbelievable stories of 2016. 1. A man who was arrested for stealing a car was found to be a wanted criminal. 2. A man who was arrested for stealing a*" |
| FUDGE | *from the world of sport. This week: a new world record for the fastest marathon, a new world record for the fastest marathon, and a new world record for the fastest marathon. The fastest marathon in the world has been set by a runner from* | *of climbing absurdity or ineptitude. Do you have a story worthy of a place in the Pit? Send it to us at [email protected] and we'll take a look! This week's story comes from* | *on Waking Times. Some of these stories are so incredible that we don't know where to begin. Everyone has their own idea of what is possible, and we hope you enjoy this collection of stories. This week's story comes* |
| ActAdd (0.1,6) | *from the world of football. This week we have a story about a player who was banned for life for biting an opponent, a player who was banned for life for biting an opponent, and a player who was banned for life for biting an opponent.* | *Every Monday we publish the most unbelievable stories of climbing stupidity submitted by our readers. See something unbelayvable? Tell us in the comments or shoot us an email at unbelayvable@climbing.com. We'll be happy to give you the* | *on Waking Times. Some of these stories are so bizarre that even we have to sit up and take notice. Others are so bizarre that we have to sit up and take notice twice. This week's story is one of the latter* |
| ActAdd (3,6) | *from the news and deeds deeds deeds deeds deeds deeds kindnesses to kindness kindnesses kindnesses kindnesses kindnesses kindness kindness kindnesss kindness ...* | *of acts of, and* | *ofpeopleeseseseses:nessesne ssesnessesnessesnesses- nesses."nessesnessesness esnessesnesses.")nessesness esnese snessesnessesnesses- nessesnessesness* |

Table K.1: Example of generation for different models and different interventions, given the same prompt, on the toxicity task.

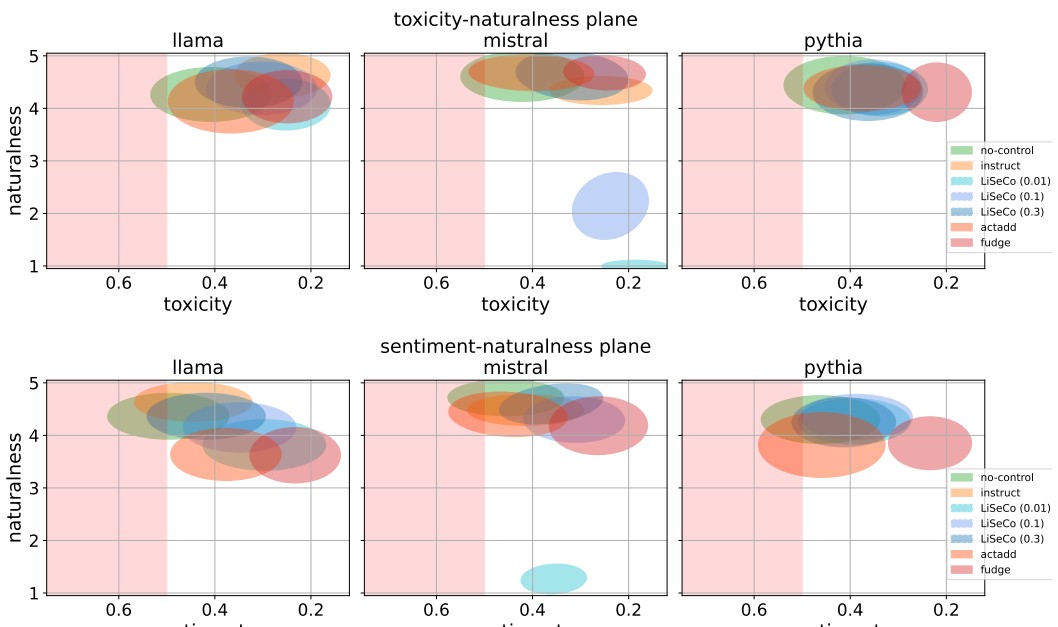

Figure L.1: **The toxicity-naturalness plane** (top) and **sentiment-naturalness plane** (bottom) for Llama, Mistral, and Pythia (left to right), on the full distribution of prompts. The **top-right corner (low toxicity, high naturalness) is best**. Each method's (toxicity/negativity, naturalness) distribution over *would-be toxic* continuations is shown as an ellipse centered at the mean, whose axes reflect $\pm 1$ SD. The red region is that labelled toxic/negative by the external classifier. LiSeCo (blue colors) shifts right, i.e., reduces toxicity/negativity, from no-control (green) and maintains high naturalness, performing on-par with instruction tuning (light orange). ActAdd (orange) least reduces toxicity/negativity. FUDGE (red), which directly optimizes w.r.t. the external classifier, most reduces toxicity/negativity as expected.

## M.2   SEMANTIC CONTROL: SENTIMENT

The sentiment score distributions for would-be toxic continuations and the full distribution are shown in Figure M.2a and b, respectively. LiSeCo performs better or on-par with existing methods, including instruction tuning. Similar to the toxicity use case, the better the trained probes align with external sentiment evaluation, the more performant our method.

**Smaller LiSeCo $p$, fewer negative generations**   Figure M.2a shows the probe and external scores for Llama, Mistral, and Pythia for the would-be negative continuations ($N = 21, 15$, and $30$), respectively. First, looking at the rows in Figure M.2a and Figure M.2b corresponding to LiSeCo, we see that LiSeCo works as expected, where decreasing $p$ thresholds the sentiment probe score to $< p$. Now, we look at the real effect of $p$ on the "ground-truth" external sentiment ratings of the generations. The intermediate rows in Figure M.2b and Figure M.2a show that, as we decrease LiSeCo $p$, the number of negative generations, as given by the external score, decreases for all models.

**Better probes, better performance**   For the sentiment task, LiSeCo performs increasingly as expected when the probe score aligns with the external score. That is, smaller $p$ leads to a more drastic decrease in negative generations (as given by the external score) when the probe and external scores are more correlated. Our method works best on Llama ($\rho = 0.27$), then Mistral ($\rho = 0.12$), both significant at $\alpha = 0.05$. Our method performs the worst on Pythia, where the correlation is insignificant ($\rho = 0.05$).

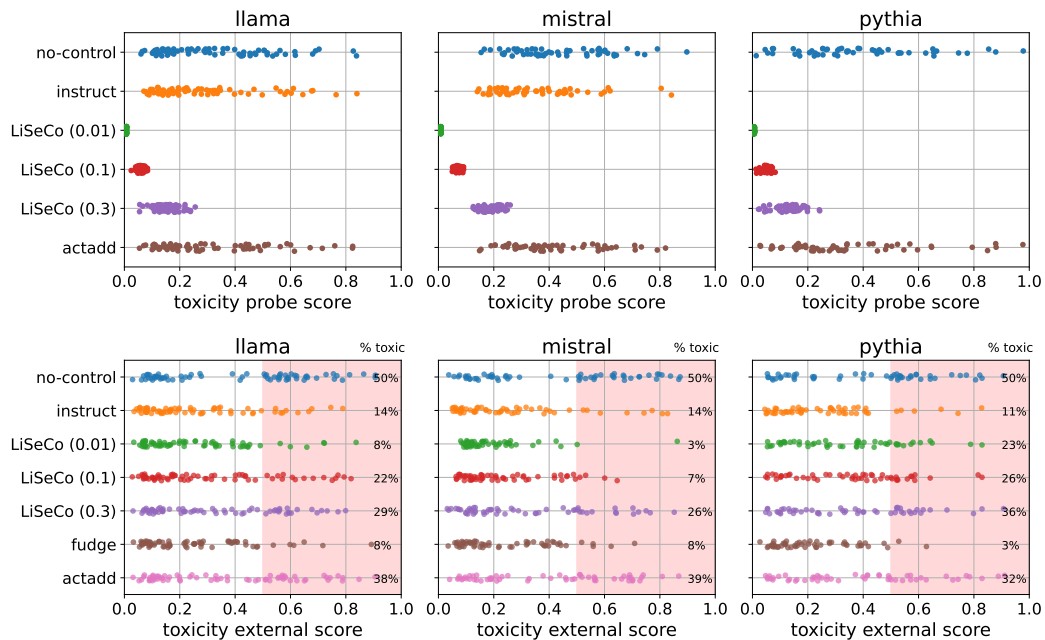

Figure M.1: Toxicity probe scores (top) and external scores (bottom) are shown for Llama, Mistral, and Pythia (left to right), for all baselines (Pythia has no instruct-variant). (Bottom) Probability for toxicity greater than $0.5$ is shaded in red, with the toxic-labeled % displayed on the right.

# N INSTRUCTIONS FOR THE HUMAN EVALUATIONS

**Experiment Instructions:**

Welcome to our experiment on evaluating text naturalness! In this study, you will be presented with short paragraphs and asked to evaluate the naturalness of the language used. Please read the instructions carefully before proceeding.

**Experiment Details:**

- You will be provided with short paragraphs of text.

- Your task is to evaluate how natural each paragraph reads. Rate it on a whole-number scale from 1 to 5, where:

  - 1 indicates the paragraph is gibberish.
  - 5 indicates the paragraph reads completely natural.

**Blind Evaluation:**

Please note that this evaluation is blind. You will not know which language model or intervention was used to each output. This ensures unbiased assessment.

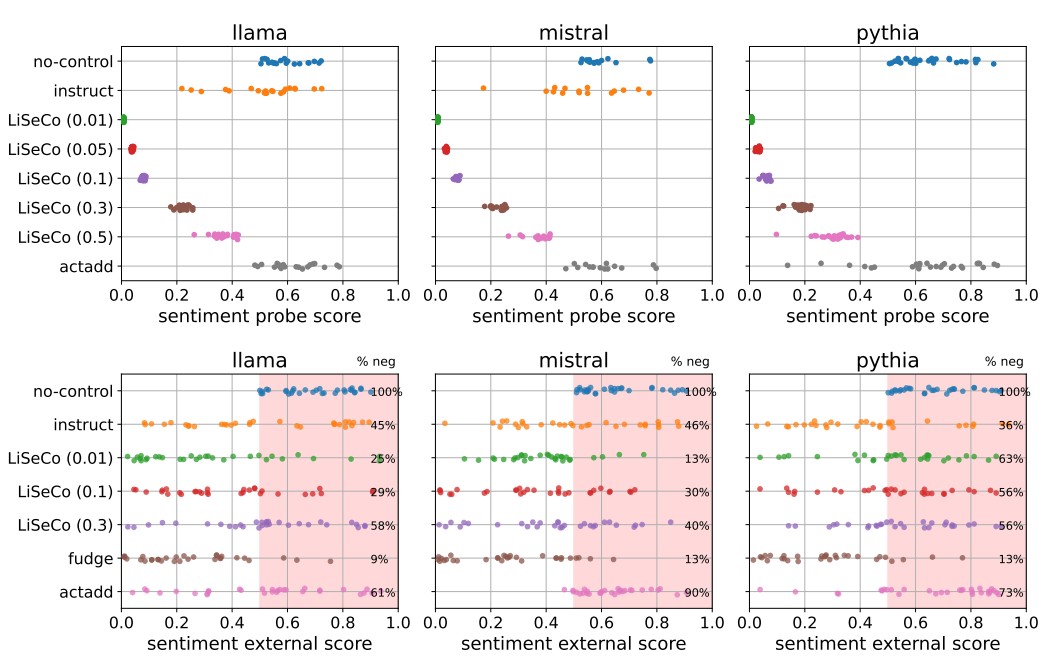

(a) Would-be negative continuations for the sentiment task. Probe scores (top) and external scores for sentiment task set (bottom), shown for all baselines and the best external score layer for ActAdd, layer 15.

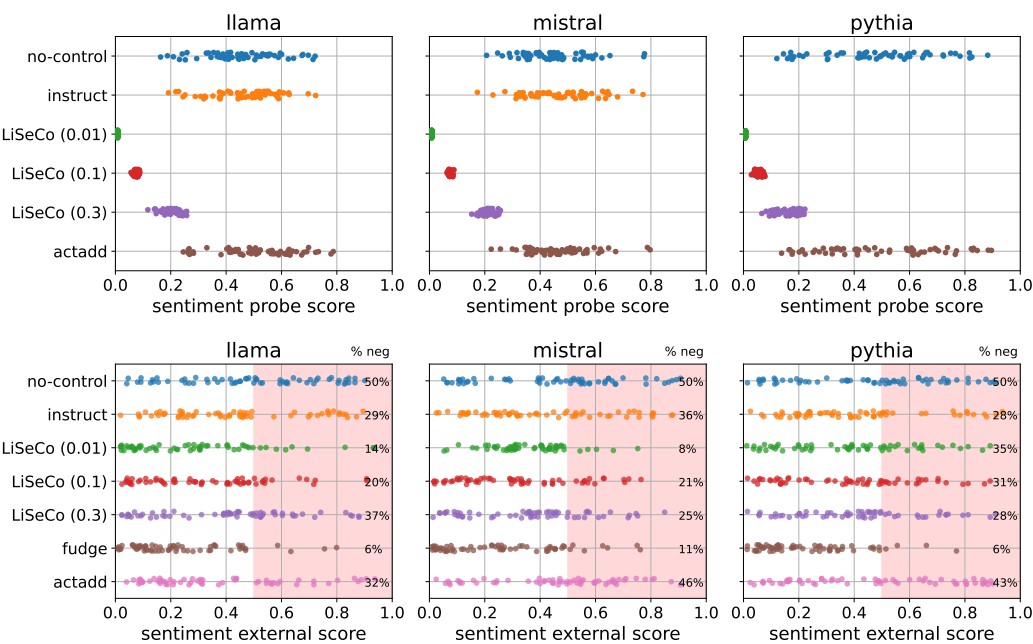

(b) Entire distribution of probe (top) and external (bottom) scores for sentiment task set ($N = 150$), shown for all baselines and the best external score layer for ActAdd, layer 15. Note that the LiSeCo $p$ parameter constrains the probe score (probability of being negative) to less than $p$.

Figure M.2: Probe score distributions for sentiment. Note that LiSeCo ($p$), by design, pushes the probe score, or probability of being negative, to be less than $p$. (a) shows the would-be negative continuations, and (b) shows the full evaluation distribution.

