# OpenReview forum: "Linearly Controlled Language Generation with Performative Guarantees"
_ICLR.cc/2025/Conference — Submitted to ICLR 2025_

### Official Review · Reviewer_2dSM · 2024-10-23

**Soundness:** 3
**Presentation:** 3
**Contribution:** 2
**Rating:** 3
**Confidence:** 4

**Summary:**

This work proposes LiSeCo, an intervention method that applies control theory principles in the latent space of language models to dynamically steer the sequence generation to desired semantic regions, such as avoiding toxic or undesired content, while preserving text quality.

**Strengths:**

1. The method is applicable in post-hoc adaptation, making it efficient and universally applicable to frozen language models.
2. The work offers a closed-form solution for optimal interventions in the latent space of language models and provides theoretical guarantees on the controllability of the method.
3. The method is designed to be lightweight and dynamic, eliminating the need for gradient-based methods, which reduces computational overhead during the generation process.

**Weaknesses:**

**Theory:**
1. *Semantic Probe:* The probing classifier function $f_t$ maps the latent space to the decision space $\[ 0, 1 \]$, meaning it assigns a single scalar value to each hidden representation (1-dimension). However, line 157 and Figure 1 suggest that the latent space is mapped to a 2-dimensional vector, as indicated by  $W_t \in \mathbb{R}^2$. This creates a discrepancy in the mathematical formulation: either the definition of $f_t$ or the dimension of $W_t$ is incorrect. In general, I assume that there is no benefit in having a second dimension. The closed-form solution could be computed as a binary classification problem (using the sigmoid instead of the softmax function) as well (see 1. Question).
2. *Optimal Controller:* In essence, you assume that $f_t$ can differentiate between toxic and non-toxic inputs. So, your control setup changes toxic inputs in a way so that $f_t$ classifies them as non-toxic. First, as part of the optimization problem, 1c and 1d are not necessary as these are no constraints and rather trivial in the context of language models (see 2. Question). Second, your control setup puts a lot of weight on $f_t$, as it ultimately assumes that $f_t$ knows how to change the hidden representations so that the language model does not generate toxic sequences.

**Evaluation:**

3. *Semantic Probe:* The probing classifier was trained on the train split of the constraint dataset, and its performance was evaluated on the corresponding validation split. The evaluation is based on sequences from the constraint dataset that may not have been generated by the language model from which the hidden representations are derived. This raises concerns about the validity of the evaluation: for instance, the language model might assign lower probabilities to toxic than to non-toxic sequences, which could have a direct influence on the hidden representations. This underlying influence, which is not accounted for in the current evaluation, could be mitigated by evaluating the performance on sequences that the language model has actually generated, using your proposed automatic scorer for determining the ground truth toxicity. This would provide a more realistic assessment of the probing classifier’s ability to separate toxic from non-toxic sequences. Since your theoretical guarantees rely on this assumption that toxic and non-toxic sequences are separable by $f_t$, this should be investigated in more detail (see 3. Question).

4. *Experiments:* It is not immediately clear how the value of the free parameter $p$ should be chosen. For instance, $p = 0.01$ seems necessary to avoid generating toxic sequences with the “llama” and “pythia” models. However, for the “mistral” model, $p = 0.3$ is the best setting, while $p = 0.01$ significantly sacrifices naturalness (see 4. Question). Moreover, for the “pythia” model, sequences are still largely classified as negative, regardless of the value of $p$. Thus, while your method theoretically provides a guarantee, this guarantee holds only if the allowable region suggested by the probing classifier aligns with the region where the language model actually generates non-toxic/non-negative sequences. The empirical results suggest that this alignment may not be as strong as claimed. If the probing classifier truly had 90% accuracy, then only slightly over 1% of sequences should have scores above 0.5 when $p = 0.01$.

**Questions:**

1. What is the benefit in having a 2-dimensional decision space?
2. What is the rational behind including 1c and 1d in the optimization problem?
3. How does the probing classifier perform on differentiating between toxic and non-toxic sequences that were actually generated by the language model?
4. Is there a clear guidance on choice of the free parameter $p$?

**Details Of Ethics Concerns:**

I don't have any ethical concerns.

---

> ### Author Response · Authors · 2024-11-20
>
> Thanks for your careful review. We’ve responded to each weakness below; please let us know if anything remains unclear or unresolved!
>
> __Semantic probe dimensionality:__
>
> We apologize for the discrepancy in the mathematical formulation, thanks for pointing this out! To clarify, the probing classifier function $f_t$​ maps the latent space to a scalar in $[0,1]$, see line 159 for the definition. Additionally, we would like to clarify that as it’s written currently, $\sigma$ refers to the softmax (line 160) mapping a 2D vector to a 2D vector; the $\sigma_2$ is equivalent to the sigmoid and denotes index 2 of the vector returned by $\sigma$. We agree that it’ll be more clear to just write $\sigma$ as the sigmoid to avoid defining $\sigma_2$ etc. We will update this in the manuscript.
>
>
> __Optimal control formulation eqs 1c,d:__
>
> Thank you for the comment! In a standard optimal control setting, constraints like 1c and 1d are generally included as they capture critical aspects of the problem: the propagation dynamics (1c) and the initial conditions (1d). While in this specific setting we relax the optimization problem, making these constraints not strictly necessary, we included them in Formulation 1 for completeness and consistency with standard optimal control problems (see Borrelli, Francesco, Alberto Bemporad, and Manfred Morari. Predictive control for linear and hybrid systems. Cambridge University Press, 2017). We would like to emphasize that 1c is particularly non-trivial. Maintaining this constraint would require performing back-propagation through the layers to find the globally optimal intervention across all layers, which is computationally intensive. By relaxing the problem and removing this constraint, we instead aim for interventions that are optimal locally at each layer. This significantly simplifies the problem while still enabling effective control. As for 1d, the initial condition directly impacts the optimal solution at the first layer, making it an essential part of the problem formulation.
>
> __Semantic probe evaluation:__
>
> Thanks for the recommendation to calibrate the LiSeCo probes on LM-generated text! We would like to highlight that all existing online intervention methods, to the best of our knowledge, use a feedback mechanism trained _not_ on LM-generated text, but rather external natural language inputs (ITI, Li 2023; Whispering Experts Suau 2023; DExperts Liu et al., 2021; ActAdd Turner et al., 2023; etc). As such, our work does not depart from the activation steering literature, and your suggestion would constitute one of the first experiments where feedback is calibrated on LLM-generated inputs. We’re currently running these experiments and will report results when we have them.
>
> Furthermore, we do already perform experiments looking at the probe and external score alignment (see Figure 3 and Appendix K), and, moreover, explicitly model the effectiveness of LiSeCo on the degree of alignment, see line 515. We show that the Spearman correlation between probe-scores and external (ground-truth scores) is higher for Llama and Mistral, models for which LiSeCo is more performant.
>
>
> __Experimental evaluation:__
>
> Thank you for raising these points. The parameter p can be seen as a toxicity “tolerance”. In terms of choosing its value, we agree that it is dependent on the model. As pointed out in the comment, the naturalness of mistral’s generations is impacted from a too stringent value of p, while for the two other models are able to handle lower values of p. This variability reflects inherent differences in the latent spaces and generative behaviors of the models. We also point out that the value of p is application-dependent. We propose for p to be treated as a degree of freedom that practitioners can tweak depending on the application. It may also be possible to learn the value of p for a given application. We will emphasize this in the paper.
>
> We agree that our provided guarantees are dependent on the classifier’s quality. We explicitly discuss this point in the Results line 515, Appendix K, Discussion lines 536-537, and propose mitigation strategies.
>
> Finally, in the main paper, we only present results constrained to would-be toxic continuations, rather than the full evaluation distribution. Results on the full evaluation distribution, consisting of 50/50 toxic and untoxic continuations, is shown in Fig K.1 for toxicity and Fig K.2 for sentiment. On the full evaluation distributions, the theoretical guarantee holds for ⅓ of the cases and especially for larger $p$.

---

> ### Comment · Reviewer_2dSM · 2024-12-01
>
> Thank you for the response.
>
> 1. Based on your clarification, it appears that Figure 1 is inconsistent.
> 2. Clarifying the optimal control formulation resolves my question.
> 3. Do you already have preliminary results? While I agree that your work "*does not depart from the activation steering literature*", it is worth noting that adhering to common approaches may not always be the most effective approach.
> 4. While the variability of $p$ across models is understandable, treating it as a “*degree of freedom*” places significant responsibility on practitioners. Additionally, even if the “*theoretical guarantee holds for ⅓ of the cases*”, the practical performance cannot be "*guaranteed*" as claimed in the paper.

---

### Official Review · Reviewer_akVP · 2024-10-25

**Soundness:** 3
**Presentation:** 3
**Contribution:** 3
**Rating:** 6
**Confidence:** 4

**Summary:**

The paper proposes to control the generation of LLMs using controls defined in their latent space. On top of these controls, the paper applies techniques borrowed from Control Theory to steer trajectories (latent representations) away from regions corresponding to undesired behaviors.

Here is a summary of their approach:

== Approach formulation:

- **Given:** We are given **an already pre-trained** LLM. The goal is to modify the internal representation of the model so that output responses: (1) are **guaranteed** to never fall into a disallowed region. (2) are as close as possible to a desired region.

- **The control mechanism:** The control mechanism alters the output vector embedding at **every layer** that guarantees (in probability) that latent trajectories remain out of the "unsafe" region. The control mechanisms here are additive, i.e., we design a controller parameter $\theta_t$ that gets added to its later embedding $x_t$, at every layer $t$.

- **Quantifying unsafe region:** The approach also requires a "probing function" that maps latent vectors to their safety/unsafety(e.g., toxicity). In practice, building this classifier requires supervision. Specifically, any embedding that satisfies $\sigma (W^\top (x_t)) \geq p$ (I).

- **Steering away from unsafe regions:** This additive control is designed so that it satisfies the following inequality   $\sigma (W^\top (\theta_t + x_t)) \leq p$ (II), for any $x_t$ that leads to "unsafe" generations characterized by Eq.I.

- **Problem formulation** Now one can formally define the goal as finding a set of controls for all layers $t$: $\{ \theta_t \}_t$, by ensuring the condition in Eq.II for each layer.

- **Simplifying the goal** Rather than solving it for **all the layers** (previous paragraph), the paper chooses a more modest goal of solving the goal for each layer. With this simplification, there is an analytical solution (Theorem 1) since the inequality of a linear separator and a nonlinearity (softmax).

**Strengths:**

- The presented formulation seems novel to me. The math is beautiful. So kudos to the authors.
 - The construction of using semantic probes for controlled generation is intuitive and interesting.
 - The closed-form solution renders the approach very efficient compared to other controlled generation approaches like FUDGE or PPLM which requires classifiers or gradients.
 - Overall writing is clean and clear. I was able to follow and understand them.
(I also have extensive feedback on wherever the writing needs to improve; see them below)

**Weaknesses:**

- I have extensive comments below where I elaborate on your writing.

- I think your experimental results are your weakness. Part of this is writing (I have elaborated on this layer). But there are other aspects too that I am unclear about which may require more experiments or better arguments.

- The work provides guarantee (in probability) that output activation will lie in the desired region. However, defining a proper "allowed region" in practice is challenging. In this work, defining allowed/disallowed regions is achieved by a linear probing classifier. Thus the performance guarantee lies on the classifier being accurate, which is not always the case. In particular, your "probing functions" are typically not very reliable on the edges of safe and unsafe generations. In other words, the guarantees that you give are on the model behavior, **assuming** that the probe is reliable (whether it is reliable or not, is not something that is quantified or guaranteed by your framework). For example, it may be difficult to probe whether early tokens in the generation would lead to unsafe responses or not because safety is usually defined on the sequence instead of token level. FUDGE takes care of this issue by training discriminators that predict whether a target attribute will become true in the future. However, it's unclear whether a simple linear prob can do this too. Moreover, probes are less effective in out-of-domain cases. It calls into question of the effectiveness of the proposed approach's OOD generalization capability, which is not sufficiently studied by the experiments.

- The framework assumes a binary allowed / disallowed region. This does not capture the need for more fine-grained control, such as the degree of toxicity or a spectrum of sentiments. This limitation is also reflected in the relatively toy experimental setup, where the labels are binary for the toxicity and negativity avoidance tasks.

-  In practice, you need to sacrifice your guarantees to maintain naturalness; so there is a clear limit on how much you can control the output safety.

 -  While you're solving layer-wise guarantees, only the guarantee on the final layer is going to matter to the user. So one is left wondering whether this layer-wise control is necessary or one can just do it for the final layer. Or perhaps apply a stronger control (higher p) for middle layers, but lower it for the final layer to maintain naturalness? These clearly require more experimental investigation.

- Unclear whether the proposed LiSeCo approach can achieve additional gains on top of instruction tuning or conventional safety alignment. Because of the lightweight nature of the proposed approach, only moderate improvements on safety can still be an helpful addition on safety-aligned models.
 - There is limited analysis of experimental results.  (a) For example, LiSeCo (0.01) outperforms FUDGE only on the Mistral model. However, Mistral+LiSeCo (0.01) shows very low naturalness (Figure 2). Maybe the output has already degraded to low quality texts. Further explanations are needed. (b) It would be interesting to conduct analysis on what is the probe’s effect on the distribution of next tokens. Is there a clear reduction of the probability of toxicity-related tokens? (c) It would be helpful to discuss the trade-off between controllability and latency a bit more (or perhaps more clearly?). It seems that LiSeCo attains worse controllability than FUDGE in many cases but is much more efficient. I have more comments on this below.

**Questions:**

Questions are listed in no particular order:

 - In the "Semantic Probe" paragraph, did you mean to define $f_t=\sigma(W_t^\top x_t)$? This seems to be implied but not explicitly given.

 - Are your control mechanisms defined for each input or for a collection of inputs? The paragraph in S3 is not explicit, but since it defines R_t for any x ($Rt = \{ x | \sigma(W^T x) > p\}$), it makes it seem like we define the control for **any** input. However, in S4, it seems like we're fixing the input. For instance, "prompt sequence" (Eq1d) and also specific choices of x_t in Theorem 1. Given the statement of your theorem, I think the latter is correct (a control per prompt), but please make it explicit upfront in S3.

- Proof of Theorem, you open by stating KKT conditions. Here you need to first define what lambda is. I suggest you reorder so that the definition of Lagrangian appears first, then define lambda, and then show the KKT condition (and say why it's important to check these for those who may not be familiar with the constrained optimization via Lagrangians)

- In Eq C.8, how did we obtain $w_1$ and $w_2$ from $W$? And from there, how do we define $w$? Seems like you're doing some reparameterizations without clearly stating it.

- [Potentially related to the previous point] The term "the difference of the columns of $W =: [W_1, W_2]$" (Theorem 1) is interesting but also a bit surprising. Intuitively why does it emerge in your solution? Also, am I interpreting  $W =: [W_1, W_2]$ correctly when we split the matrix into two vertical matrices? (i.e., we don't imply that $W$ has only two columns).

- In your optimization formulation in Eq1, you're missing additional term(s) that specify the domain of $x_t$ and their membership to the unsafe region: $\sigma(W^\top x_t) > p$ (not just any point) similar to the condition in Eq. 3a.

- While you're solving layer-wise guarantees, only the guarantee on the final layer is going to matter to the user. I think you want to articulate it clearly. Perhaps the only rationale for intermediate layer adaptation is to simplify the achievement of safety in the final layer, which is a hypothesis that needs to be verified.

> We consider instruction-tuning, which relies on extensive LM finetuning, to be a target, or “upperbound", baseline.

Why are these considered as an "upperbounds"? If you're using the same set of supervision used for your reward models, the direct comparison to these supervised/tuned models is a fair game.

- Have you considered trying other additive tuning approaches, say, LoRA or BitFit? They're other additive adaptations that I'm forgetting here.

> FUDGE has the potential to incur a high inference latency compared to other methods
>Of the methods tested, FUDGE has the highest latency by several orders of magnitude, around 3 seconds compared to other baseline

But these k sequences can be parallelized since they're independent of each other; right?

> This comes at the benefit of directly optimizing for the evaluator. For this reason, we consider FUDGE to be an upper-bound baseline, given that other steering methods do not have access to the ground-truth scorer.

Your approach is also using the probe's feedback; so not sure if I buy this. If I am missing something please elaborate.

> The fact that probes attain high accuracies of ∼90% confirms that the disallowed toxic (negative) regions Rt are linearly decodable with high probability

This is misleading. Given that you apply your intervention per layer, one needs to look at the layer-wise accuracy. And we see that 85% holds typically for layer ten or higher. Instead of Table 1, I'd actually include your figures in the appendix which, IMO, are easier to parse.

- One odd thing about Fig2 is that, upon applying intervention, "naturalness" (fluency) gets higher than "no-control" (the green box); for example, top-right subfigure for llama. Given that the base model (no intervention) is optimized for fluency, this is clearly signals noise in your evaluation.

- Consider categorizing your baselines into two: (1) tuning-base (training, LoRA, etc.); (2) inference-time steering (FUDGE, yours, etc.). And right off the bat, you want to say that category (1) is not easily extensible. The discussion of category (2) is tricky. Currently, your arguments rely on the slow speed of FUDGE, but that’s a weak argument (FUDGE can be implemented faster, I think, as I mentioned above). Plus, there are (many?) other adaptation-based that you do not compare with. I think your strengths here are: (1) your proposal is principled and allows for guarantees (even though, in practice, you need to sacrifice this guarantee; otherwise, you miss out on naturalness, which defeats the whole purpose); (2) it requires producing distribution over the words in the vocabulary. Yours is just an embedding projection.

> modifying LM weights and can be extremely energy-intensive, **needing > 4 orders of magnitude more data**

I don’t think you show any result on this (comparison with tuning LM, as a function of its supervision size).

- It’s worth viewing your EQ1 in terms of (and comparing against) other existing known frameworks, such as SVM https://en.wikipedia.org/wiki/Support_vector_machine

- Lastly, I think the “control-theoretic” framing of the paper might be a bit of a stretch. Ultimately, Eq1 is the most general form of your problem and Eq2 is what you actually solve. If KKT condition and interpretation as the Bellman equation constitute a “control paper,” all the alignment papers that also use some variant of RL or sequence prediction (and are most of AI now) are also “control papers.” I acknowledge that it’s a bit murky what constitutes a “control” paper; so I won’t hold this against you.

=== Minor

- Fig2: in the caption, I’d say what 0.01, 0.1, … in the figure are (I think they’re $p$). I’d even revise the legend to say `LiSeCo(p= … )` for absolute clarity

- I don't see where you define $\mathcal{R}_t^C$ (just under the Theorem). Previously, we only defined $\mathcal{R}_t$.

 - Expressions such as $p(unsafe) < p$ (under Theorem 1) are confusing. I suggest you choose a different notation for probability $P(.)$ so that it does not get confused with your design parameter $p$.

 - In various places there is a subscript $2$ attached to $\sigma$ (the softmax function). Unsure what that means. I don't think it's a standard notation.

- Figure 2 and 3 are not colorblind-friendly and colors are Figure 2 color is hard to distinguish.

> Activation Addition (ActAdd) Turner et al. (2023)

Use `\citep{}`

= Missing citations:

There is a ton of literature on controlled generation (https://arxiv.org/abs/2202.11705, https://arxiv.org/abs/2205.14217, https://arxiv.org/abs/2112.08726, https://openreview.net/forum?id=kTy7bbm-4I4) that try to solve a similar problem. I have included a few below, but there is a ton that I am missing.
As far as I know, these works do not provide guarantees that you do, but I don’t think it’s helpful for our collective progress to ignore them.

---

> ### Author Response · Authors · 2024-11-20
>
> First, we are super grateful that you wrote such a thoughtful and thorough review of our paper, and that you appreciated the math. We’re splitting our response into separate comments on different themes. We hope to have addressed most of your questions, and we’ll definitely be implementing the text changes you suggested. Please let us know if anything remains unclear.

---

> > ### Author Response · Authors · 2024-11-20
> > **LiSeCo design and theoretical properties**
> >
> > __The term "the difference of the columns W=:[W1,W2]" (Theorem 1) is interesting but also a bit surprising. Intuitively why does it emerge in your solution?__
> >
> > Good question. The vector $w_1 - w_2$ is the normal vector to the separating hyperplane learned in logistic regression. Then, Theorem 1 says the optimal solution is some constant * that vector. This essentially moves $x_t$ to the hyperplane.
> >
> > To see why $w_t$ is the normal vector, we can quickly derive it:
> >
> > 1. The separating hyperplane is all $x$ s.t. $$p(toxic) = \sigma_2 (W^\top x) = \frac{e^{w_2^\top x}}{e^{w_1^\top x} + e^{w_2^\top x}} = 0.5,$$
> > Where $\sigma_2$ is the 2nd index of the softmax (or sigmoid).
> >
> > 2. Rearranging some terms, the separating hyperplane is all $x$ s.t.
> > $$(w_1 - w_2)^\top x = 0.$$
> >
> > Now, we can see that $w_1 - w_2$ emerges as the normal vector to the separating hyperplane.
> >
> > __OOD generalization capability__
> >
> > We agree that the guarantees are for the "oracle" linear probe, which should be approximated as closely as possible in practice. We discuss this issue in the limitations. In order to evaluate the practical implications of this limitation, we perform experiments looking at the probe and external score alignment (see Figure 3). Although dependent on the model, our experiments show that our method behaves well in practice for external scores as well.
> >
> > __Binary classification of the region:__
> >
> > Thanks for this comment. We interpreted your point two different ways:
> >
> > 1. Why binary classification into a safe/unsafe region? Because the decision to intervene or not is inherently binary. Even if we want to tune toxicity continuously along some hypothetical range, e.g. “we want our sequence to have `toxicity=0.1’”; or “we want our sequence to have toxicity in a neighborhood $0.1 \pm 0.01$”, there needs to first be a check (classification) whether or not our latent already satisfies this condition.
> >
> > 2. If we want to do “continuous tuning” instead of intervening based on discrete categories, note that Theorem 1 actually returns the same optimal solution for “ensure toxicity <= 0.1” (our current formulation) and “set toxicity=0.1” (continuous tuning), where the only change is turning an inequality constraint to an equality constraint. This can be verified by plugging in the current solution 3a and seeing that the resulting probability in 2b is equal to p.
> >
> > We’ll definitely discuss this after presenting the Theorem.
> >
> > __Tradeoff between performance and naturalness:__ While it is true that in some cases a higher tolerance (value of p) lead to better naturalness, this was not the case for other models in a reasonable range of $p$ (see llama and pythia). The tradeoff is model and classifier dependent, and in some cases the tradeoff seems to not be strong. We also highlight that other SOTA methods do not achieve neither guarantees on performance nor on naturalness.
> >
> > __Multi-layer vs last-layer intervention:__ We have explored the idea of intervening only at the last layer, as well as interventions on the layer where the classifier exhibits the highest accuracy score (for instance, in llama this is layer 16). However, these single-layer interventions were not as effective. This is due to the sequential nature of the problem, where activations travel through layers in a temporal order. When actuation starts early on, i.e. in the first layers, this leads to a smoother overall intervention that allows to preserve naturalness. An analogy would be braking a car before hitting an obstacle: it is more effective and less disruptive to the dynamics to start braking early on smoothly than braking strongly right before the obstacle. We agree that exploring different values of $p$ at the different layers would be a very relevant study that could potentially lead to a boost in naturalness while maintaining the effectiveness. We defer a theoretical analysis of the optimal distribution of p across layers to future work.
> >
> > __It’s worth viewing your EQ1 in terms of (and comparing against) other existing known frameworks, such as SVM https://en.wikipedia.org/wiki/Support_vector_machine__
> >
> > Thanks for this comment. When designing LiSeCo we had also considered linear SVM. Ultimately, we chose logistic regression over SVM as it has a nice probabilistic formulation (which is the core of our guarantee), while the backbone of SVM is geometric instead of probabilistic, relying on maximal separation of categories. There is definitely room for further theoretical work on SVM, and what guarantees it can provide; we’ll add to Discussion.

---

> > > ### Author Response · Authors · 2024-11-20
> > > **Experiments**
> > >
> > > __LiSeCo in addition to instruction tuning:__
> > > Thanks for the suggestion. We have not looked into how LiSeCo can be combined with instruction tuning. We are currently implementing experiments on this and will post the results when ready. LiSeCo+Instruct will definitely be in the revised version.
> > >
> > > __Limited experimental analysis:  (a) For example, LiSeCo (0.01) outperforms FUDGE only on the Mistral model. However, Mistral+LiSeCo (0.01) shows very low naturalness (Figure 2). Maybe the output has already degraded to low quality texts.__
> > >
> > > Indeed the output had degraded to low quality texts. See for instance the center column of Table I.1 for an example. We'll discuss this further in the Results.
> > >
> > > __(b) It would be interesting to conduct analysis on what is the probe’s effect on the distribution of next tokens. Is there a clear reduction of the probability of toxicity-related tokens?__
> > >
> > > We’re investigating and we’ll let you know what we find.
> > >
> > > __One odd thing about Fig2 is that, upon applying intervention, "naturalness" (fluency) gets higher than "no-control" (the green box); for example, top-right subfigure for llama. Given that the base model (no intervention) is optimized for fluency, this is clearly signals noise in your evaluation.__
> > >
> > > This is a great observation. We found that models without control tended to get stuck in limit cycles, i.e. produce degenerate repetitive inputs when sampled (row 1 of Table I.1); we discuss this in the footnote in line 431 and Appendix I. Instead, we think the intervention "kicked" the language generation out of the repetitive loop. This kind of repetitive behavior in LMs is well-documented in the literature [1,2]. We’ll add this discussion to the results.
> > >
> > > [1] [The Curious Case of Neural Text Degeneration, Holtzman et al., 2020](https://arxiv.org/abs/1904.09751) \
> > > [2] Learning to Break the Loop: Analyzing and Mitigating Repetitions for Neural Text Generation, Xu et al., 2023.

---

> > > > ### Author Response · Authors · 2024-11-20
> > > > **Baselines**
> > > >
> > > > __Instruction-tuning as “upperbound" baseline: Why are these considered as an "upperbounds"? If you're using the same set of supervision used for your reward models, the direct comparison to these supervised/tuned models is a fair game… modifying LM weights and can be extremely energy-intensive, needing > 4 orders of magnitude more data. I don’t think you show any result on this (comparison with tuning LM, as a function of its supervision size).__
> > > >
> > > > Due to compute constraints, we did not finetune any LM from scratch, but rather used the SOTA Instruct-LMs directly from the company page on Huggingface. The >4 orders of magnitude more data alludes to the token count from the AI@Meta Huggingface page, see https://huggingface.co/meta-llama/Llama-3.1-8B-Instruct#training-data . To make this clear, we will directly state that the Llama-Instruct we use is aligned using 25M tokens + publicly available datasets (it is unclear from the Mistral paper how much data was used to train their Instruct model). We considered these to be upper-bound baselines as we thought it unrealistic that a lightweight, low sample complexity intervention like LiSeCo would outperform large-scale instruction tuning/RLHF.  Surprisingly, LiSeCo performed as well as Instruct in the negativity avoidance task.
> > > >
> > > > __Have you considered trying other additive tuning approaches, say, LoRA or BitFit? They're other additive adaptations that I'm forgetting here.__
> > > >
> > > > Thanks for the suggestion; we defer testing these baselines to future work, but will cite them in Related Work.
> > > >
> > > > __FUDGE has the highest latency by several orders of magnitude, around 3 seconds compared to other baseline. But these k sequences can be parallelized since they're independent of each other; right?__
> > > >
> > > > You’re right that $k$ sequences can be parallelized. Because of memory constraints, we had to run FUDGE (as well as all other baselines) with a batch size of 1. An off-the-shelf calculation: (additional latency FUDGE - base latency) / additional latency LiSeCo, tells us that $k \approx 350$ sequences would have to be parallelized in one batch, for FUDGE’s latency to match the other baselines. We’ll add this to the discussion.
> > > >
> > > > __This comes at the benefit of directly optimizing for the evaluator. For this reason, we consider FUDGE to be an upper-bound baseline, given that other steering methods do not have access to the ground-truth scorer. Your approach is also using the probe's feedback; so not sure if I buy this. If I am missing something please elaborate.__
> > > >
> > > > We'll clarify. We evaluate the toxicity of an LM generation using a neural scorer that takes a sequence -> [0, 1].
> > > >
> > > > In FUDGE, they optimize the generated output token wrt a neural scorer: sequence -> [0, 1]. Using the score, they recalibrate the token probabilities.
> > > >
> > > > We used the same neural toxicity scorer for evaluating all baselines, and also as the neural scorer in the FUDGE pipeline. So FUDGE outputs the optimal logits against our evaluation metric, hence it is an upper bound on performance-- we included it precisely for this upper-bound comparison.

---

> > > > > ### Author Response · Authors · 2024-11-20
> > > > > **Framing**
> > > > >
> > > > > __It would be helpful to discuss the trade-off between controllability and latency a bit more (or perhaps more clearly?). It seems that LiSeCo attains worse controllability than FUDGE in many cases but is much more efficient.__
> > > > >
> > > > > Thanks for the suggestion– indeed it’s interesting to view online intervention methods as a tradeoff within the triplet (performance, test-time compute, train-time compute), where test-time latency, in our case, proxies test-time compute.
> > > > >
> > > > > For  FUDGE, the test-time compute is higher than LiSeCo, which also buys FUDGE a higher performance. Instruct, on the other hand, requires high train-time compute, and 0 additional test-time compute, with decent performance; ActAdd requires neither test-time nor train-time compute, to the detriment of performance.
> > > > >
> > > > > We think this view casts different methods in a fairer light; rather than just optimizing performance, we can see clearly the tradeoffs between performance and resource constraints that depend on the practitioner.
> > > > >
> > > > > __Consider categorizing your baselines into two: (1) tuning-base (training, LoRA, etc.); (2) inference-time steering (FUDGE, yours, etc.). And right off the bat, you want to say that category (1) is not easily extensible.__
> > > > >
> > > > > Thanks so much for this; it’s a great idea. We’ll frame the baselines accordingly.
> > > > >
> > > > > __the “control-theoretic” framing of the paper might be a bit of a stretch.__
> > > > >
> > > > > At the core of optimal control is intervening on a system’s output without changing the system itself. As such, LiSeCo (+ other activation steering methods), which computes an online additive intervention without fundamentally changing the system weights, is a control approach. On the other hand, finetuning and knowledge editing are interventions that change the weights of the system, and lose the dynamical view inherent to control theory. Still, although LiSeCo has a strong control framing, we don’t call the paper a control paper :)

---

> > > > > > ### Author Response · Authors · 2024-11-20
> > > > > > **Presentation / writing clarifications**
> > > > > >
> > > > > > __Given that you apply your intervention per layer, one needs to look at the layer-wise accuracy. And we see that 85% holds typically for layer ten or higher. Instead of Table 1, I'd actually include your figures in the appendix which, IMO, are easier to parse.__
> > > > > >
> > > > > > Thanks for the suggestion; we agree that it’s more transparent to have the Appendix figures directly in the main. We had compromised for space reasons by putting Table 1, but we’ll find a way to move the figures.
> > > > > >
> > > > > > __In the "Semantic Probe" paragraph, did you mean to define ft=σ(Wt⊤xt)? This seems to be implied but not explicitly given.__
> > > > > >
> > > > > > Yes! It is formally defined in the Semantic Probe paragraph line 160.
> > > > > >
> > > > > > __I think the latter is correct (a control per prompt), but please make it explicit upfront in S3.__
> > > > > >
> > > > > > Exactly– thanks for the suggestion– we will add a line in Section 3.2 (Approach) in the first paragraph.
> > > > > >
> > > > > > __Proof of Theorem, you open by stating KKT conditions. Here you need to first define what lambda is. I suggest you reorder so that the definition of Lagrangian appears first, then define lambda, and then show the KKT condition (and say why it's important to check these for those who may not be familiar with the constrained optimization via Lagrangians)__
> > > > > >
> > > > > > Thanks for this, we agree it’ll read more smoothly. We’ll add the definition of the Lagrangian first, define $\lambda$, then apply the KKT conditions as you suggested.
> > > > > >
> > > > > > __In Eq C.8, how did we obtain w1 and w2 from W? And from there, how do we define w? Seems like you're doing some reparameterizations without clearly stating it.__
> > > > > >
> > > > > > Sorry about that– we define $w_t := W_t^1 - W_t^2$ in Theorem 1 (line 259). $W_t^i$ is the $i$th column of $W_t \in \mathbb R^{d\times 2}$, where $W_t$ has two columns mapping $x_t$ to $\mathbb R^2$ as we’re doing binary classification. We will restate Theorem 1 at the top of Appendix C so that there’s notation continuity.
> > > > > >
> > > > > > __Fig2: in the caption, I’d say what 0.01, 0.1, … in the figure are (I think they’re p). I’d even revise the legend to say LiSeCo(p= … ) for absolute clarity__
> > > > > > Thanks! We’ll make this change.
> > > > > >
> > > > > > __I don't see where you define RtC (just under the Theorem). Previously, we only defined Rt.__
> > > > > > Sorry, by $\mathcal R_t^C$ we mean the set-complement of $\mathcal R_t$. We’ll also say the definition in words rather than just introducing the $C$.
> > > > > >
> > > > > > __Expressions such as p(unsafe)<p (under Theorem 1) are confusing. I suggest you choose different notation for probability P(.) so that it does not get confused with your design parameter p.__
> > > > > >
> > > > > > Good point, we’ll use $\pi(\cdot)$ or $f_t(\cdot)$ for the probability.
> > > > > >
> > > > > > __In various places there is a subscript 2 attached to σ (the softmax function). Unsure what that means. I don't think it's a standard notation.__
> > > > > > Here, we’re indexing into the 2nd element of the $\sigma$ softmax function, which in our case maps $\mathbb R^2 \to \mathbb R^2$. Since this caused some confusion for another reviewer, we’re planning to get rid of softmax altogether and just use the sigmoid (which is also commonly denoted $\sigma$ unfortunately), whose range is in $[0, 1]$.
> > > > > >
> > > > > > __Figure 2 and 3 are not colorblind-friendly and colors are Figure 2 color is hard to distinguish.__
> > > > > >
> > > > > > Sorry about that, we’ll change the color palette to a colorblind-friendly one.
> > > > > >
> > > > > > __Missing citations__ We’ll add these to the paper, thanks so much!

---

### Official Review · Reviewer_JNbS · 2024-11-03

**Soundness:** 3
**Presentation:** 3
**Contribution:** 2
**Rating:** 5
**Confidence:** 3

**Summary:**

This paper presents a novel approach, Linear Semantic Control (LiSeCo), for controlled language generation within Large Language Models (LLMs). The method employs a control-theoretic approach, optimizing latent space interventions to guide LLM outputs away from undesired regions (which are found by probing), particularly demonstrated for toxicity avoidance. Unlike other approaches that rely on complex or computationally heavy adjustments, LiSeCo operates with low latency through a closed-form solution and provides probabilistic performance guarantees.

**Strengths:**

1. This paper is well written, easy to understand and the proposed method is not complicated, it just intervenes on the hidden representations.

2. The proposed method is empirically effective without inducing much computational overhead.

3. The empirical results are backed up with theoretical guarantees.

**Weaknesses:**

1. The paper mentions that small modifications in latent space can have unpredictable outcomes (similar to the "butterfly effect"), yet there are no experiments analyzing such unintended consequences in practice. A detailed analysis on potential adverse effects of these interventions, especially when they involve semantically sensitive regions of latent space, would lend weight to the discussion and address the potential risks in practical applications. It would be crucial to see that the proposed method maintains / minimally decreases the quality of the text.

2. The applicability - there are certain attributes that are probably more easily probed (e.g. in this work toxicity - since this is associated with some of the toxic words). But there are also attributes that are not (e.g. tone, formality…). Extending this control to more nuanced tasks would really show the effectiveness / limitation of this approach.

If the author can resolve my concerns here, I would be happy to raise my score.

**Questions:**

035 - 040: There are also decoding time algorithms for controllable generations, which are omitted in this paragraph.

042:  For instance, while knowledge editing provides an efficient alternative to exhaustive retraining, it poses risks akin to the butterfly effect: minor adjustments could lead to unintended consequences - this is not backed up with citations.

---

> ### Author Response · Authors · 2024-11-20
>
> Thanks so much for your review! We hope to have responded to the weaknesses below. The first weakness we think is a writing clarification point, and the second we intend to resolve with experiments. _Please let us know whether we've resolved your concerns!_
>
>
> __The paper mentions that small modifications in latent space can have unpredictable outcomes (similar to the "butterfly effect"), yet there are no experiments analyzing such unintended consequences in practice__
>
> Thanks for the comment! We will highlight more carefully that this statement in the Introduction applies specifically to knowledge editing in LMs, which are methods that update LM weights , e.g. ROME (Meng et al., 2022), (not the latent space, like LiSeCo). Our “butterfly effect” statement is a paraphrase of an existing argument in the literature (Cohen et al., 2023; Pinter et al., 2023). This argument states that, when editing/updating facts in LM weights, like “The new UK Prime Minister is Keir Starmer”, it may not be true that this fact will get propagated correctly across the LM’s knowledge graph. Again, we meant for the “butterfly effect” statement to only apply to knowledge editing and not other methods. We apologize for the confusion.
>
>
> __042: butterfly effect: minor adjustments could lead to unintended consequences - this is not backed up with citations.__
>
> Thanks for pointing this out; that was our oversight. We’re adding the relevant citations:
> Evaluating the Ripple Effects of Knowledge Editing in Language Models (Cohen et al., TACL 2023)
> Emptying the Ocean with a Spoon: Should We Edit Models? (Pinter et al., EMNLP 2023)
>
>
> __It would be crucial to see that the proposed method maintains / minimally decreases the quality of the text.__
>
> We evaluate text quality using LM perplexity and human naturalness ratings; see lines 369-371. The dependence of naturalness on LiSeCo (p) (and other baselines) is shown in Figure 2. As expected, the more extreme the intervention, the less natural the outputted text.
>
>
> __Extending this control to more nuanced tasks would really show the effectiveness / limitation of this approach.__
>
> Thanks for the suggestion; we are currently extending our experiments to formality steering as you recommend. If you have any recommendations for formality datasets, please let us know! Stay tuned for those results.
>
> 035 - 040: There are also decoding time algorithms for controllable generations, which are omitted in this paragraph.
> Thanks for the pointer! We address these algorithms in lines 071-75 in Related Work, but will also add decoding-time algorithms to lines 035-40 in the Introduction.

---

> > ### Comment · Reviewer_JNbS · 2024-11-24
> > **Reviewer Response**
> >
> > I think the authors have partially resolved my concerns. However, one of my main concern about this (applicability to other characteristics other than toxicity) is not resolved. Therefore, I will maintain my score.

---

> > > ### Author Response · Authors · 2024-11-25
> > >
> > > Hi, thanks for your response and patience. As stated, we’re currently working on formality results which should land today.
> > >
> > >  Note also that our experiments are on _both toxicity and negativity reduction_ — these are both in the main paper (see Fig 2) but for space reasons we prioritized toxicity in the main and negativity in the appendix.

---

### Official Review · Reviewer_ucXH · 2024-11-06

**Soundness:** 3
**Presentation:** 3
**Contribution:** 3
**Rating:** 6
**Confidence:** 3

**Summary:**

The paper proposed an interesting, gradient-free method to control the language generation. The authors introduce a method called LiSeCo, where they first train layer-specific probers on data within the "unsafe" region and then apply these probers at inference using a closed-form solution.

**Strengths:**

1. The paper is well organized and clearly presented.
2. The author used an interesting locally-optimized close-form solution to significantly speed up the inference process.
3. The performance of semantic control and quality maintenance looks good.

**Weaknesses:**

1.  The tasks selected by the authors appear somewhat too old and easy to me. It would be beneficial to see the method applied to more recent and challenging tasks, particularly those related to safety. For example, datasets like UltraSafety (https://huggingface.co/datasets/openbmb/UltraSafety) or HarmfulQA (https://huggingface.co/datasets/declare-lab/HarmfulQA) could provide a better evaluation of the method's effectiveness in handling harmful or unsafe content. Tasks like IMDB reviews are too straightforward for current large language models and may not adequately showcase the capabilities of the proposed approach.

2. The evaluation could be significantly improved. The authors rely on automatic evaluations using relatively small models (such as RoBERTa-based models) and self-assessments. This approach may not offer a comprehensive understanding of the method's performance. A common practice is to employ more advanced models like GPT-4 for evaluation, especially since the models discussed in the paper are around 7B parameters. Utilizing GPT-4 could provide a more robust and thorough assessment.

3. The colors in Figure 2 is hard to read but it is ok.

**Questions:**

N/A

---

> ### Author Response · Authors · 2024-11-20
>
> Thanks so much for your careful review. We hope to have responded to the weaknesses you raised below. Please let us know which weaknesses remain unresolved, if any!
>
> __The tasks selected by the authors appear somewhat too old and easy to me. It would be beneficial to see the method applied to more recent and challenging tasks, particularly those related to safety.__
>
> Thanks for this comment. Unfortunately, while these objectives aren’t very new in NLP, toxicity and negativity mitigation are still not solved in the general case– this is shown in our experimental results and in the recent literature: our exact datasets, like the Jigsaw toxicity dataset which includes subtle forms of toxicity (Adams et al., 2017), are still widely used in new model steering papers (Rodriguez et al., 2024; Suau et al., 2023; inter alia).
>
> In response to your and Reviewer 2’s suggestion to test a more challenging/nuanced task, we’re currently extending experiments to control text formality. While not UltraSafety or HarmfulQA, we think a challenging task in a completely new domain would be appreciated. We will post results once we have them!
>
>
> __Tasks like IMDB reviews are too straightforward__
>
> We do not task LMs to reduce negativity on IMDb reviews; IMDb just forms part of the dataset used to calibrate the linear probes. We constructed the probe-training sets to be general and high-coverage, with the goal to train probes on a representative sample of the LMs’ latent space. Towards this, the IMDb dataset you mention was just a small part of a diverse compilation of datasets used to train sentiment probes.
>
> __evaluation could be significantly improved. The authors rely on automatic evaluations using relatively small models (such as RoBERTa-based models) and self-assessments.__
>
> Thanks also for this suggestion. While the RoBERTa-based models are smaller than, e.g., GPT4, we note that they are specifically finetuned on toxicity and sentiment classification, achieve SOTA or near-SOTA accuracy on benchmarks, are open-source, and are up-to-date as of 2022 (TimeLM, Loureiro et al., 2022).
>
> We collected human ratings (authors + another lab member), which is the gold standard text evaluation method. But, we emphasize that they are not _self-assessments_, as we took extreme care that the authors who generated the LLM outputs didn’t rate them. This is an important point we’ll emphasize in the Methods.
>
> Thank you for your suggestion to use GPT4 for evaluation; it indeed is an option for scaling up. In our experiments, we were hesitant to use LLMs like GPT4 to produce toxicity scores, which for us are continuous likelihood estimates in [0, 1]. This is because LLM-as-judge is quite recent in the literature, and we’re currently unaware of work validating Claude or GPT4 for similar continuous ratings (Zheng et al., 2023 evaluates LLM-as-judge on mostly categorical data). Since we had the bandwidth to collect manual naturalness scores, we did not consider automating that either. Still, for the new experiments on formality, we’ll pilot using GPT or Claude for evaluation and report how that goes.
>
> [1] Judging LLM-as-a-Judge with MT-Bench and Chatbot Arena, Zheng et al., 2023.

---

> > ### Comment · Reviewer_ucXH · 2024-11-20
> >
> > I thank the authors for their response. I do not have further questions and will keep my score. Good luck :)

---

> > > ### Author Response · Authors · 2024-11-20
> > >
> > > Thanks for your quick turnaround :)

---

### Author Response · Authors · 2024-11-28
**Experimental and theory update**

Our current work experiments on contrained generation for two concepts: _toxicity and negativity._

We extended our approach to account for features that can be continuously tuned. Our theory now tackles cases where, instead of eliminating a specific feature on a text, it keeps it within a range, or controls the degree to which a specific property is present/missing. This enables more controllability on the generation. We refer to this extension as "continuous tuning" in the manuscript. We provide the theoretical results and mathematical proofs that ensure guarantee satisfaction in Appendix E.

We also provide preliminary experimental results that demonstrate the effectiveness of this extension through a formality use-case, where the degree of formality is set implicitly by LiSeCo $p$. Hence, we find this to be a good use-case to test the validity of our method at controlling the degree of a specific feature (in this case formality), as opposed to eliminating it completely. Our experimental setup and results are reported in Appendix F. We find that LiSeCo, better than other baselines, is able to vary the formality without sacrificing text quality.

Please let us know if anything remains unresolved about the theoretical and experimental extension to our work.

---

> ### Author Response · Authors · 2024-11-28
> **Writing update**
>
> In the new version, the vast majority of writing concerns have been resolved (R3) and we've normed the terminology to $\sigma$ being the sigmoid (R4). We believe that it reads clearer now.

---

### Meta-Review · Area_Chair_HJED · 2024-12-22

**Metareview:**

This paper proposes a method for controllable text generation. The approach first trains a linear probe to classify attributes from hidden states, and then derives a closed-form solution for perturbing those states to ensure that the predicted attribute probabilities are within a certain thresholds. Experiments on toxicity and sentiment control demonstrate this method.

Strengths:
1. The method has a closed-form solution and only requires training the probes, and is therefore efficient.

Weaknesses:
1. Reviewers raised a concern that this work only conducted experiments on toxicity and sentiment control and is not shown on more applications such as safety in language models.

Overall, the idea proposed in this paper is very interesting and the proposed solution is simple and efficient. However, reviewers were concerned about whether the proposed method works in broader applications such as safety in language models. Therefore, I'm recommending rejection for the current paper, but I wouldn't mind if the paper gets accepted.

**Additional Comments On Reviewer Discussion:**

See above. reviewers were mainly concerned that the current experiments were done on toxicity and sentiment control but not more recent tasks such as safety. Authors mentioned extending experiments to more tasks but seem to not have provided new results on them.

---

### Decision · Program_Chairs · 2025-01-22

Reject